# Long first exons and epigenetic marks distinguish conserved pachytene piRNA clusters from other mammalian genes

Tianxiong Yu [1,2,7], Kaili Fan [1,2,7], Deniz M. Özata [3], Gen Zhang[4], Yu Fu [2,5,6], William E. Theurkauf[4✉], Phillip D. Zamore [3✉] & Zhiping Weng [1,2✉]

In the male germ cells of placental mammals, 26–30-nt-long PIWI-interacting RNAs (piR-NAs) emerge when spermatocytes enter the pachytene phase of meiosis. In mice, pachytene piRNAs derive from ~100 discrete autosomal loci that produce canonical RNA polymerase II transcripts. These piRNA clusters bear 5′ caps and 3′ poly(A) tails, and often contain introns that are removed before nuclear export and processing into piRNAs. What marks pachytene piRNA clusters to produce piRNAs, and what confines their expression to the germline? We report that an unusually long first exon (≥ 10 kb) or a long, unspliced transcript correlates with germline-specific transcription and piRNA production. Our integrative analysis of tran-scriptome, piRNA, and epigenome datasets across multiple species reveals that a long first exon is an evolutionarily conserved feature of pachytene piRNA clusters. Furthermore, a highly methylated promoter, often containing a low or intermediate level of CG dinucleotides, correlates with germline expression and somatic silencing of pachytene piRNA clusters. Pachytene piRNA precursor transcripts bind THOC1 and THOC2, THO complex subunits known to promote transcriptional elongation and mRNA nuclear export. Together, these features may explain why the major sources of pachytene piRNA clusters specifically gen-erate these unique small RNAs in the male germline of placental mammals.

[1]Department of Thoracic Surgery, Clinical Translational Research Center, Shanghai Pulmonary Hospital, The School of Life Sciences and Technology, Tongji University, 200092 Shanghai, China. [2]Program in Bioinformatics and Integrative Biology, University of Massachusetts Medical School, Worcester, MA 01605, USA. [3]RNA Therapeutics Institute, University of Massachusetts Medical School, Worcester, MA 01605, USA. [4]Program in Molecular Medicine, University of Massachusetts Medical School, Worcester, MA 01605, USA. [5]Bioinformatics Program, Boston University, 44 Cummington Mall, Boston, MA 02215, USA. [6]Present address: Oncology Drug Discovery Unit, Takeda Pharmaceuticals, Cambridge, MA 02139, USA. [7]These authors contributed equally: Tianxiong Yu, Kaili Fan. ✉email: william.theurkauf@umassmed.edu; phillip.zamore@umassmed.edu; zhiping.weng@umassmed.edu

In animal germ cells, 21–35 nt long PIWI-interacting RNAs (piRNAs) guide PIWI proteins to repress transposons and regulate gene expression[1]. Eutherian mammals produce three distinct types of germline piRNAs during spermatogenesis. In mice, piRNAs first appear in the fetal testis; these piRNAs initiate transposon silencing via DNA methylation[2–5]. A second set of piRNAs, pre-pachytene piRNAs, emerges in the neonatal testis[2]. The function of pre-pachytene piRNAs, which typically derive from mRNAs, remains unknown. Finally, when spermatocytes enter the pachytene phase of meiosis, male mice produce a third type of piRNAs: pachytene piRNAs[6–9]. Unique to placental mammals, the 26–30 nt long pachytene piRNAs derive from ~100 discrete autosomal loci that contain fewer transposon sequences than the genome as a whole[7,10–12]. Pachytene piRNAs are both highly diverse—a single spermatocyte produces hundreds of thousands of different piRNA species—and highly abundant, reaching a peak intracellular concentration (7.2 μM) that rivals the abundance of ribosomes[13]. Yet the biological functions and target RNAs of pachytene piRNAs remain to be defined. Some studies suggest that pachytene piRNAs silence mRNAs during spermiogenesis[14–18], while others support the view that they have no sequence-specific function[19].

The genomic regions that produce piRNAs, referred to as piRNA clusters or piRNA source loci, resemble canonical mRNA- and long noncoding RNA- (lncRNA) producing genes: they are transcribed by RNA polymerase II (RNA pol II); bear 5′ caps and 3′ poly(A) tails; their transcription start sites (TSS) are marked with histone H3 trimethylated on lysine 4 (H3K4me3); and their transcripts often contain introns that are removed before nuclear export and processing into piRNAs[11]. Mouse pachytene piRNAs derive from 100 well-defined loci. An additional 84 annotated loci generate pre-pachytene piRNAs, while 30 "hybrid" loci produce piRNAs with characteristics from both classes[11]. Human adult and juvenile testes produce piRNAs from at least 83 pre-pachytene, 10 hybrid, and 89 pachytene loci[12]. In both mice and humans, the pachytene piRNA clusters and the genes encoding the proteins required for piRNA production are coordinately activated by the transcription factor A-MYB, which is first expressed as spermatogonia enter meiosis I[11,12]. A-MYB also activates transcription of other genes required for meiosis, but not implicated in piRNA biogenesis or function[20].

Like many genes transcribed in the germline during spermatogenesis, the promoters of pachytene piRNA clusters contain fewer CG dinucleotides than protein-coding genes expressed in the soma, and most of the CGs are cytosine methylated[21], a DNA modification normally associated with silencing rather than active transcription. These atypical promoters tend to be inactive in the soma. High levels of 5-hydroxymethylcytosine (5hmC) and active histone marks, such as H3K4me3 and histone H3 acetylated on lysine 9 (H3K9ac) or lysine 27 (H3K27ac) have been proposed to facilitate expression of pachytene piRNA clusters in mouse and human testis[21]. In mice, roughly half of pachytene piRNA clusters bind the transcription elongation factor BTBD18, and *Btbd18* mutant male mice are sterile and show decreased transcription and piRNA production from these loci[22].

Among the 100 well-annotated mouse pachytene piRNA clusters[11], 41 are divergently transcribed from a central promoter. These include 15 pairs that produce piRNAs from both arms, 7 that produce piRNAs from one arm and an mRNA from the other, and 4 that produce piRNAs from one arm and a lncRNA from the other. Similarly, among the 89 well-annotated human pachytene piRNA clusters[12], 46 are divergently transcribed from a central promoter, including 18 pairs that produce piRNA precursors from both arms, 7 that produce an mRNA from the other arm, and 3 that generate a lncRNA from the other arm. What genic or epigenetic features direct the transcripts of pachytene piRNA clusters into the piRNA pathway, what determines which arm of these divergently transcribed loci makes piRNAs, and what prevents the expression of pachytene piRNAs outside the testis remain unknown.

Here, we report that an unusually long (≥10 kb) first exon or long unspliced transcript correlates with germline-specific production of piRNA precursor transcripts from mouse pachytene piRNA clusters. We further report that pachytene piRNA precursors from long-first-exon or long unspliced transcripts are preferentially bound by THOC1 and THOC2, subunits of the THO complex, which is required for transcription elongation and nuclear export of mRNAs[23–25]. In *Drosophila*, the THO complex is essential for piRNA biogenesis, and *tho* mutants are female sterile[26–28]. Our integrative analysis of transcriptome, piRNA, and epigenome datasets across multiple species reveals that a long first exon is an evolutionarily conserved feature of pachytene piRNA clusters. Finally, comparison of testis germ cells with a variety of somatic tissues and cell types suggests that a highly methylated promoter, often containing a low or intermediate level of CG dinucleotides, correlates with germline expression and somatic silencing of pachytene piRNA clusters.

## Results

**Pachytene piRNA clusters are expressed in the testis but silent in somatic tissues.** In placental mammals, mature piRNAs are primarily detected in gonads[6,7,9,29,30]. Our analysis of transcriptome data[31] shows that the primary precursor transcripts of pachytene piRNAs, but not pre-pachytene or hybrid piRNAs, are also restricted to male gonads. Pachytene piRNA precursor transcripts were abundant in the adult mouse testis, but not in adult brain, colon, liver, skeletal muscle, heart, kidney, lung, or spleen. In contrast, the precursor transcripts of pre-pachytene and hybrid piRNAs, which often also function as mRNAs, were found in all tissues examined (Fig. 1a). Like human fetal ovaries[30], mouse fetal ovaries (embryonic day 11.5–20.5) behave like somatic tissues: most pachytene piRNA clusters are not expressed, whereas hybrid and pre-pachytene piRNA clusters are actively transcribed (Fig. 1a). Among the 100 mouse pachytene piRNA clusters, transcripts from 69 were detected solely in the testis—in the other tissues we examined, these transcripts were present at <0.1 RPKM (reads per kilobase of transcript per million mapped reads). The steady-state transcript abundance of another 21 pachytene piRNA clusters was ≥4-fold greater in the testis than in any of the eight somatic tissues analyzed. Of the remaining ten pachytene piRNA clusters, eight—two encoding proteins and six producing lncRNAs—were more abundant in the testis than in other tissues, but by <4-fold, while two additional loci were more abundant in the soma than in the germline. Thus, 90% of annotated mouse pachytene piRNA clusters are testis specific. In contrast, just 13% of hybrid (4/30) piRNA clusters and only one of the 84 well-defined pre-pachytene piRNA clusters are testis specific.

On average, piRNAs from the 90 testis-specific pachytene piRNA clusters were 4.2-fold more abundant than from the ten pachytene piRNA clusters with broader expression (Wilcoxon rank-sum test $p$ value = 0.036). Consistent with their testis-specific expression, pachytene piRNA clusters showed higher RNA pol II occupancy in the testis than in somatic tissues. Active histone marks—the promoter mark H3K4me3; the enhancer marks H3K4me1, H3K4me2, and H3K27ac; and the transcriptional elongation mark H3K36me3—were also substantially higher for these 90 loci in the testis than the soma (Wilcoxon signed-rank test $p$ value ≤ $3.0 \times 10^{-10}$; Fig. 1b). The signal profiles at individual piRNA clusters (Supplementary Fig. 1 shows H3K27ac) reveal that the enrichment of these histone marks is

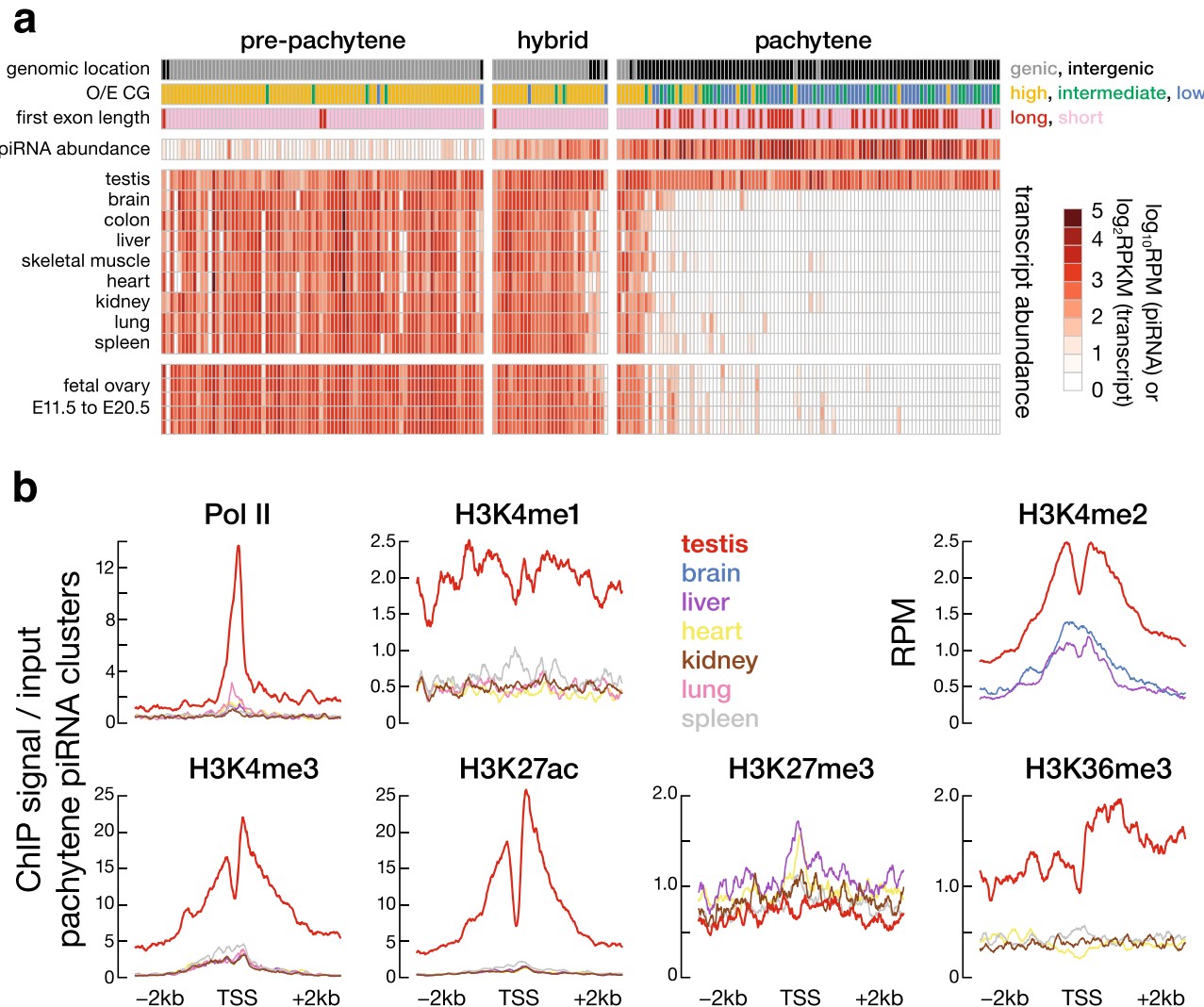

**Fig. 1 Mouse pachytene piRNA clusters show testis-specific expression and epigenetic signals. a** Each column represents a piRNA gene, grouped by type (pachytene, hybrid, and pre-pachytene) and sorted by tissue-specificity score ("Methods") within each group. Each piRNA gene is further annotated by its function as an mRNA (black/grey), low, intermediate, or high-CG level at the promoter (low-CG, intermediate-CG, or high-CG; in blue, green, and yellow), length of first exon or length of intronless gene (≥10 or <10 kb; in red and pink), piRNA abundance (RPM) in adult testis, and transcript expression level (RPKM) in the testis, eight somatic tissues, and fetal ovary. piRNA abundance and transcript expression levels are $\log_2$ and $\log_{10}$ transformed, respectively, with a pseudo-count of 1 and share the same color scale. **b** Each panel indicates the levels of RNA pol II binding or a histone mark; tissues are color coded. Pachytene piRNA clusters exhibit high levels of activating histone marks (H3K4me3, H3K4me2, H3K4me1, H3K27ac, and H3K36me3), low levels of the repressive histone mark H3K27me3, and high levels of RNA pol II binding specifically in adult testis. ChIP-seq signals are shown in a ±2 kb window around the TSS in 10 bp bins. H3K4me2 ChIP-seq data were publicly available only for testis, liver, and brain. We performed two-sided Wilcoxon signed-rank tests to compare epigenetic signals at pachytene piRNA clusters in the testis versus somatic tissues. Pol II, $p < 2.2 \times 10^{-16}$; H3K4me1, $2.3 \times 10^{-16} < p < 6.9 \times 10^{-16}$; H3K4me2, $1.0 \times 10^{-11} < p < 3.0 \times 10^{-10}$; H3K4me3, $p < 2.2 \times 10^{-16}$; H3K27ac, $p < 2.2 \times 10^{-16}$; H3K27me3, $2.3 \times 10^{-11} < p < 7.9 \times 10^{-6}$; H3K36me3, $1.7 \times 10^{-16} < p < 2.4 \times 10^{-16}$.

across most piRNA clusters. We note that several mutually exclusive histone marks are enriched at piRNA clusters, such as H3K4me1, H3K4me2, and H3K4me3; acetylation and butylation; and 5mC and 5hmC (also see below). We presume that the marks are likely to exist in different subsets of cells. Conversely, pachytene piRNA clusters had 1.3–2.0-fold lower levels of the repressive histone mark H3K27me3 in the testis than in somatic tissues (Wilcoxon signed-rank test $p$ value ≤ $7.9 \times 10^{-6}$; Fig. 1b).

Consistent with their expression profiles (Fig. 1a), three hybrid pachytene piRNA clusters showed testis-specific profiles of histone marks, while the remaining hybrid pachytene piRNA clusters had high levels of active histone marks in multiple somatic tissues (Supplementary Fig. 2a). As expected from their broad germline and somatic expression, pre-pachytene piRNA

clusters on average showed similar profiles of histone marks in testis and somatic tissues (Supplementary Fig. 2b).

For comparison with pachytene piRNA clusters, we identified a set of 1,171 testis-specific, protein-coding genes with a steady-state transcript abundance ≥10 RPKM in the testis and ≥4-fold greater expression in testis than any of the eight reference somatic tissues. Like pachytene piRNA clusters, testis-specific protein-coding genes had substantially greater RNA pol II binding, higher levels of active histone marks, and lower levels of the repressive histone mark H3K27me3 in adult testis than in somatic tissues (Wilcoxon signed-rank test $p$ value < $2.2 \times 10^{-16}$; Supplementary Fig. 2c).

In mice and other placental mammals, a single protein, A-MYB, activates transcription of pachytene piRNA clusters and a

large fraction of genes encoding piRNA pathway proteins[11,12]. Mice have 17 such annotated piRNA pathway genes, and the steady-state mRNA abundance was higher than 10 RPKM in the testis for all 17 genes; furthermore, 15 of the 17 were expressed at levels ≥4-fold greater in testis than in somatic tissues (Supplementary Fig. 3a). Of the remaining two genes, *Ddx39* (DExD-box helicase 39 A) was expressed in all tissues, while *Mov10l1* (Mov10-like RISC complex RNA helicase 1) was also expressed in the heart, where it was originally discovered[32]. Consistent with their high expression in testis, the 15 genes had substantially higher levels of active histone marks and RNA pol II binding, and significantly lower levels of the repressive histone mark H3K27me3 in adult testis compared with the panel of somatic tissues (Supplementary Fig. 3b). Most piRNA pathway genes are expressed in mouse fetal ovaries (Supplementary Fig. 3a), consistent with the activities of the piRNA pathway in the female gonads in mouse and human[3,29,30,33–35].

Sixty nine of the 100 annotated pachytene piRNA clusters, but only one of the 17 piRNA pathway genes, were exclusively expressed in the testis (<0.1 RPKM in all somatic tissues examined; Chi-square test *p* value = $3.9 \times 10^{-5}$). Thus, most pachytene piRNA clusters are expressed exclusively or specifically in the mouse testis, and generally show greater tissue specificity than genes encoding piRNA pathway proteins.

**Long first exon and low CG correlate with piRNA production**. Pachytene piRNA clusters display two unusual features: long first exons (long is defined as ≥10 kb and short as <10 kb throughout) and low level of CG dinucleotides (defined in "Methods") at their promoters (Fig. 1a), features known to repress somatic expression of protein-coding genes[36–38]. We asked whether these two features had a similar relationship to germline expression of pachytene piRNA clusters. Because most precursor transcripts of piRNA clusters are processed into mature piRNAs, we used piRNA abundance as a surrogate for the expression levels of piRNA clusters. Our data show that for pachytene piRNA clusters, long first exons and low-CG promoters positively correlate with piRNA abundance (Fig. 2a).

Most protein-coding and lncRNA genes have short exons, including first exons (median length of annotated first exons = 229 nt); a long first exon delays splicing and hinders transcriptional elongation[36,37]. In contrast, half of mouse pachytene piRNA clusters are either long and intronless (*n* = 39; median gene length = 31,826 nt) or possess a long first exon (*n* = 8; median first exon length = 39,772 nt). Moreover, for pachytene piRNA clusters, first-exon length (or gene length for unspliced genes) correlates with piRNA abundance (*ρ* = 0.63 in pachytene spermatocytes; Fig. 2a; *p* value < $2.2 \times 10^{-16}$). Among the 21,956 annotated protein-coding and 3496 lncRNA-producing genes in mice, all have short first exons (Fig. 2b) except for three protein-coding genes that have both long- and short-first-exon isoforms and one lncRNA gene with multiple long-first-exon isoforms. However, none of these long-first-exon isoforms is expressed in the testis. No hybrid and only three pre-pachytene piRNA clusters have long-first-exon isoforms (i.e., supported by RNA-seq reads; Chi-square test between pachytene and other piRNA clusters *p* value = $6.8 \times 10^{-14}$). Furthermore, for two of these pre-pachytene piRNA clusters, the long-first-exon isoforms produced more piRNAs than the short-first-exon isoforms (piRNA density was 15.8 vs. 2.6 RPKM for *pi-Ccrn4l.1* and 12.2 vs. 0.1 RPKM for *pi-Phf20.1* in postnatal day 10.5 testes). The third pre-pachytene piRNA cluster, *7-qD2-40.1*, only makes a long-first-exon isoform, and it makes abundant piRNAs (242.6 RPKM in postnatal day 10.5 testes). Together, our data indicate that a long first exon (or a long unspliced transcript) is a specific feature of pachytene

piRNA clusters, which may distinguish them from protein-coding and lncRNA genes.

Paradoxically, high levels of promoter DNA methylation, which would typically repress the expression of protein-coding genes, do not impede expression of pachytene piRNA clusters in the testis: pachytene piRNA clusters with high piRNA abundance tend to have low promoter CG and high promoter methylation. More than 80% of pachytene piRNA clusters have promoters with low CG (defined as the ratio of the observed count over the expected count of CG dinucleotides, or O/E CG < 0.25, see "Methods"; 49/100 loci) or intermediate CG (0.25 < O/E CG < 0.5; 33/100 loci), whereas the promoters of most pre-pachytene (78/84) or hybrid (26/30) piRNA clusters are high CG (O/E CG > 0.5; Fig. 2c; Chi-square test *p* value < $2.2 \times 10^{-16}$ for high-CG vs. low-CG and intermediate-CG combined). In somatic cells, the promoter O/E CG of a protein-coding gene is anticorrelated with its methylation level and correlated with its expression level: low-CG promoters are typically methylated and repressed, while high-CG promoters are unmethylated and expressed[39]. Similarly, for protein-coding genes in the germline, promoter O/E CG correlates with expression level (Spearman correlation coefficient *ρ* = 0.68 in the forebrain and *ρ* = 0.54 in pachytene spermatocytes, *p* values < $2.2 \times 10^{-16}$), while both promoter O/E CG and expression level anticorrelate with promoter methylation level (*ρ* = −0.70 and −0.56 in the forebrain; *ρ* = −0.83 and −0.49 in pachytene spermatocytes; all *p* values < $2.2 \times 10^{-16}$). In pachytene spermatocytes, promoter O/E CG and methylation level of pachytene piRNA clusters are similarly anticorrelated (*ρ* = −0.81; *p* value < $2.2 \times 10^{-16}$); however, expression anticorrelates with promoter O/E CG (*ρ* = −0.31; *p* value = $1.5 \times 10^{-3}$) and correlates with promoter methylation level (*ρ* = 0.21; *p* value = $8.7 \times 10^{-4}$; Fig. 2a). We note that the level of promoter methylation of most genes, including pachytene piRNA clusters, remains constant throughout mouse spermatogenesis[21].

**A broad domain of histone acylation decorates long-first-exon and long unspliced pachytene piRNA clusters**. Despite their high levels of promoter DNA methylation, pachytene piRNA clusters are efficiently transcribed and generate abundant piRNAs in the testis. What additional features of the pachytene piRNA-producing loci allow them to be expressed in the testis, while other genes with highly methylated promoters are repressed? Among the various chromatin and RNA pol II signatures measured in mouse male germ cells, promoter lysine acylation—including acetylation, butyrylation, and crotonylation—correlated with the abundance of mature piRNAs (*ρ* = 0.49–0.66, *p* values ≤ $2.1 \times 10^{-7}$; Fig. 2a). Acylation of histone lysine residues activate transcription by opening chromatin, and allowing transcription factors and the transcriptional machinery to access the DNA[40–45]. The promoters of pachytene piRNA clusters displayed higher acylation levels than other classes of piRNA clusters, genes encoding piRNA pathway proteins, A-MYB-bound protein-coding genes, or testis-specific protein-coding or lncRNA genes (Fig. 2d, left panels, median ratio = 1.43–3.04, Wilcoxon rank-sum *p* values < $2.2 \times 10^{-16}$ with respect to each of the other gene sets). Moreover, many pachytene piRNA clusters were broadly marked with acylated histones, extending far into the gene body (Fig. 2d, right panels; we note that the apparent enrichment of signals upstream of pachytene piRNA cluster is due to the large number of bidirectionally transcribed piRNA clusters, and this apparent enrichment is not observed for the RIP signal described below that is specific to the transcribed strand). In contrast, other active genes showed a narrow peak of histone acylation at the transcriptional start site. These data suggest that such broad domains of acylated histones enable testis-specific transcription of

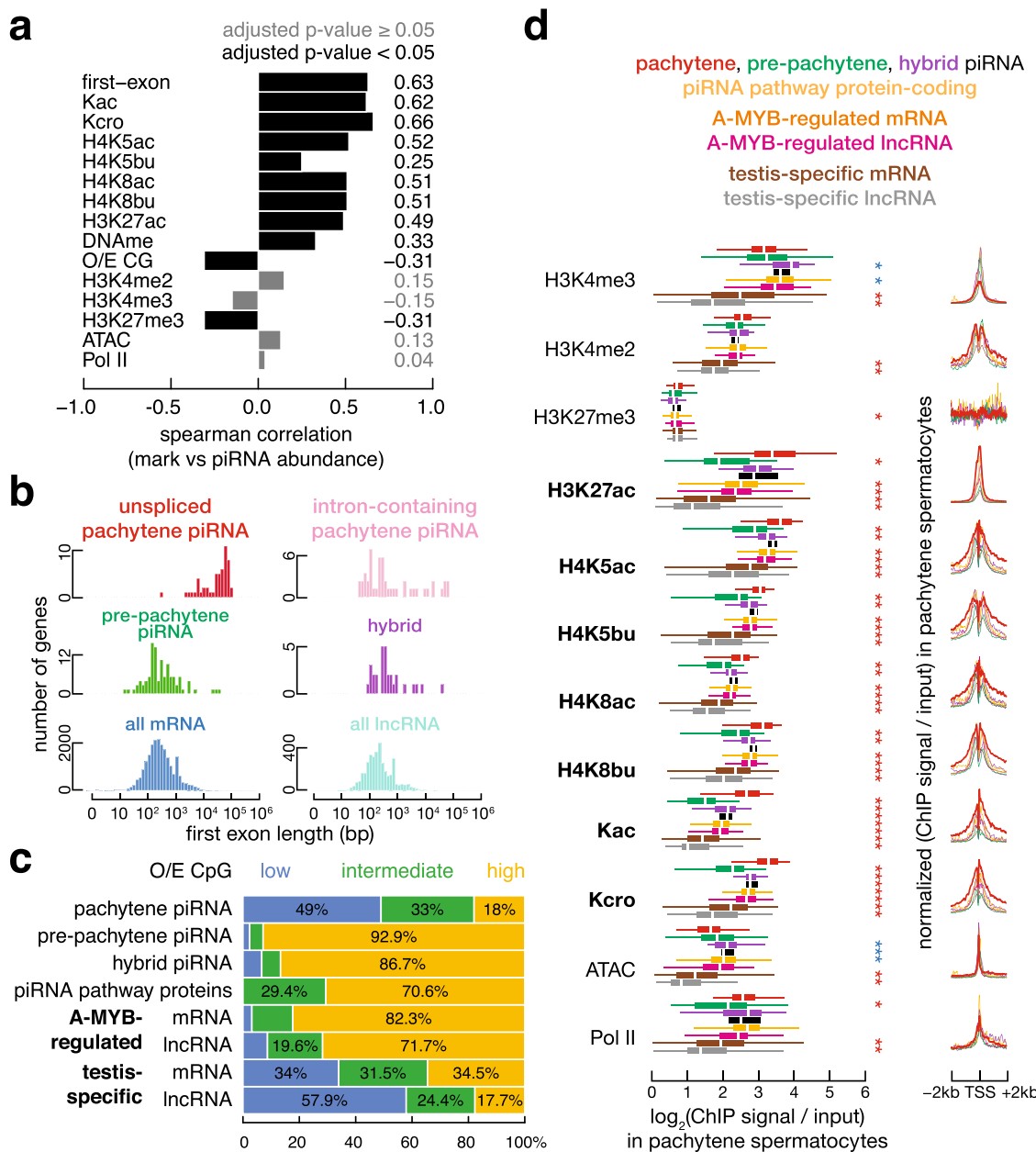

**Fig. 2 Low-CG promoter, long-first-exon or long intronless gene, and high histone acylation correlate with the production of pachytene piRNAs. a** Bar plot reports the Spearman correlation coefficient between piRNA abundance and first exon length and promoter O/E CG, H3K4me3, H3K27me3, H3K4me2, H3K27ac, H4K5ac, H4K5bu, H4K8ac, H4K8bu, pan-lysine acetylation (Kac), pan-lysine crotonylation (Kcro), chromatin accessibility measured by ATAC-seq, DNA methylation, and RNA pol II binding at the promoters (TSS ± 2 kb) of pachytene piRNA clusters in pachytene spermatocytes. Significant correlations (Benjamin-adjusted $p$ values < 0.05; $n = 100$) are marked in black, while nonsignificant correlations are marked in grey. **b** Histograms of first exon length of intronless pachytene piRNA clusters and the first exons of intron-containing pachytene piRNA clusters, pre-pachytene piRNA clusters, hybrid piRNA clusters, all protein-coding genes, and all lncRNAs. **c** The percentages of low-CG, intermediate-CG, and high-CG promoters. Eight groups are shown: pachytene piRNA clusters, pre-pachytene piRNA clusters, hybrid piRNA clusters, piRNA pathway genes, A-MYB-regulated protein-coding genes, A-MYB-regulated lncRNA genes, testis-specific protein-coding genes, and testis-specific lncRNA genes. **d**. Boxplots and meta-gene plots show histone modifications, ATAC, and RNA pol II levels at 100 pachytene piRNA clusters, 84 pre-pachytene piRNA clusters, 30 hybrid piRNA clusters, 17 piRNA pathway genes, 789 A-MYB-regulated protein-coding genes, 46 A-MYB-regulated lncRNAs, 1171 testis-specific protein-coding genes, and 164 testis-specific lncRNAs in pachytene spermatocytes. The $x$-axis for the box and the $y$-axis for the meta-gene plots report $\log_2$ ChIP signal or ATAC-seq read coverage relative to input, using a pseudo-count of 1. Asterisks indicate statistical significance (two-sided Wilcoxon rank-sum test $p$ value < 0.001) for pairwise comparisons between 100 pachytene piRNA clusters and each of the other gene types. Red (blue) indicates that the epigenetic level is significantly higher (lower) in pachytene piRNA clusters. For boxplots, whiskers show 95% confidence intervals, boxes represent the first and third quartiles, and the vertical midline is the median.

pachytene piRNA clusters despite their high levels of promoter DNA methylation.

Supporting this view, only long unspliced or long-first-exon pachytene piRNA clusters displayed high levels of histone acylation across a broad domain. Among the 100 annotated mouse pachytene piRNA clusters, 47 make long (i.e., >10,000 nt) transcripts: 39 of the 52 unspliced genes have transcripts longer than 10,000 nt; 5 of the 45 intron-containing loci have long first exons; two genes produce both long unspliced and long-first-exon transcripts from the same locus; and a single locus, *pi-1700016M24Rik.1*, generates one long unspliced transcript and two short-first-exon transcripts. The histone acylation profiles of these 47 pachytene piRNA clusters differ markedly from the 53 that make short unspliced or short-first-exon transcripts. Long unspliced or long-first-exon pachytene piRNA clusters show a broad domain of histone acylation extending far downstream from the promoter, often continuing for more than 40,000 nucleotides. In contrast, histone acylation of short intronless or short-first-exon pachytene piRNA clusters is confined to the promoter region, a pattern typical for actively transcribed mRNA- and lncRNA-producing genes (Fig. 3a–d, and Supplementary Figs. 4 and 5). Unlike acylation, other histone marks (e.g., H3K4me3), as well as most regions of high chromatin accessibility (ATAC-seq), were confined to the promoter for pachytene piRNA-producing loci (Fig. 3c, d and Supplementary Fig. 5a, b). Notably, among the 47 long unspliced or long-first-exon pachytene piRNA clusters, the density of histone acylation across the first exon or the entire intronless gene, rather than the density across the promoter, best correlated with piRNA abundance (e.g., $\rho = 0.78$ and $p$ value $< 2.2 \times 10^{-16}$ for lysine acetylation at any position of the histone tail; Supplementary Fig. 4c). Together, our data suggest that a broad domain of histone acylation across either the first exon or the entire body of intronless genes, explains the testis-specific transcription of nearly half the annotated pachytene piRNA clusters.

**BTBD18 is essential for transcription of long unspliced or long-first-exon pachytene piRNA clusters**. A subset of pachytene piRNA clusters require the nuclear protein BTBD18 to generate high levels of precursor transcripts and piRNAs[22]. BTBD18 binds the promoters of 51 pachytene piRNA clusters, and for 48 of these loci, the steady-state levels of both precursor transcripts and piRNAs decrease ≥2-fold in *Btbd18*[Null] homozygous testes, compared with heterozygous littermates. These 48 BTBD18-dependent pachytene piRNA-producing loci include 46 of the 47 pachytene piRNA clusters that produce long transcripts—39 long intronless genes, 5 long-first-exon genes, and 2 genes producing both long intronless and long-first-exon transcript isoforms (Fig. 3e).

Two additional BTBD18-dependent pachytene piRNA-producing loci have first exons shorter than 10,000 nt, but nonetheless longer than typical: the first exon of *7-qD2-11976.1* is 9436 bp and the first exon of *7-qD1-654.1* is 4620 bp. Another exception is *pi-1700016M24Rik.1*, the only pachytene piRNA gene that makes both a long unspliced transcript and two spliced transcripts with short first exons; BTBD18 binds the *pi-1700016M24Rik.1* promotor (Supplementary Fig. 5c). This locus failed to qualify as BTBD18-dependent because in *Btbd18*[Null] homozygous testis, exonic reads increased 12% while intronic reads decreased 33%. Yet piRNAs mapping to the locus decrease 16-fold in *Btbd18*[Null] mutant testis. In *Btbd18* heterozygotes, 85% of piRNAs from the locus map to introns, suggesting that the piRNAs derive from the long, unspliced isoform. Supporting this idea, intronic piRNAs decreased 17-fold in *Btbd18*[Null] testis. Just 4.6% of *pi-1700016M24Rik.1* piRNAs mapped to sequences unique to the two short-first-exon transcript isoforms, and even these piRNAs

decreased only 4.3-fold in *Btbd18*[Null] testes. Together, these data suggest that BTBD18 is essential for transcription of long unspliced and long-first-exon pachytene piRNA clusters.

Chromatin immunoprecipitation (ChIP)-seq data[22] show that BTBD18 binds the promoter of all 47 unspliced or long-first-exon pachytene piRNA clusters and the two BTBD18-dependent pachytene piRNA clusters, whose long first exons were slightly shorter than the 10 kb cutoff we imposed (*7-qD2-11976.1* and *7-qD1-654.1*). For 33 of the 49 genes, BTBD18 binding extends at least to the midpoint of the first exon (Supplementary Fig. 4a; relative extension index ≥ 0.5, see "Methods"). Similarly, for 32 of the 49 genes, the ATAC-seq signal extends at least halfway across the first exon or gene body (Supplementary Fig. 4a). Both the BTBD18 and the ATAC-seq signal extends to or beyond the midpoint of the first exon or intronless gene body for more than half of the 49 pachytene piRNA clusters (25/49). Together, these data support the proposal that BTBD18 facilitates transcription of pachytene piRNA clusters by increasing chromatin accessibility[22].

Twenty six of the 49 pachytene piRNA clusters form divergent pairs transcribed from a common, bidirectional promoter. Four additional BTBD18-dependent pachytene piRNA clusters are also transcribed from a central promoter, in which the other member of the pair does not produce piRNAs: three are paired with protein-coding genes, and one is paired with a lncRNA. For example, the BTBD18-dependent pachytene piRNA gene *10-qB4-6488.1* is transcribed from a 475-bp long, bidirectional promoter that generates the *Ddx50* mRNA from the opposite arm (Supplementary Fig. 5d). Both BTBD18 and A-MYB bind this divergently transcribed promoter. The ChIP-seq signal of BTBD18 is broad and covers the entire promoter, while the ChIP-seq peak of A-MYB is equidistant from the two TSSs. Yet only the piRNA-producing arm requires BTBD18 and A-MYB for its transcription (Supplementary Fig. 5d). Notably, *10-qB4-6488.1* makes three piRNA-producing transcript isoforms: a long unspliced RNA and two long-first-exon RNAs. In contrast, *Ddx50* has a short first exon and makes no piRNAs. Since these two genes share a promoter, their distinct transcriptional regulation and the contrasting fates of their transcripts likely reflects their different first-exon lengths and not BTBD18 binding per se.

**THOC1 and THOC2 specifically bind transcripts from long unspliced and long-first-exon pachytene piRNA clusters**. The THO complex, which acts in transcription elongation and mRNA nuclear export[23–25,46–48], also participates in piRNA biogenesis in *Drosophila*[26–28]. In the fly ovary, THO complex components specifically bind piRNA precursor transcripts, and loss of THO complex subunits disrupts transposon silencing, reduces germline piRNA abundance, and leads to male and female sterility. Our data suggest that in adult mouse testis, the THO complex also binds pachytene piRNA precursor transcripts.

We immunoprecipitated two mouse THO subunits—THOC1 (also named HPR1) and THOC2—from whole adult testes and sequenced the co-immunoprecipitated RNA (RIP-seq); immunoprecipitation with IgG served as a control (Supplementary Data 1). THOC1 and THOC2 specifically associated with long unspliced and long-first-exon transcripts from pachytene piRNA clusters: on average, binding of pachytene piRNA precursor transcripts to THOC1 and THOC2 was 7.2 and 5.5 times greater than spliced mRNAs, respectively (Fig. 4a; Wilcoxon rank-sum test $p$ value $< 2.2 \times 10^{-16}$). In contrast, THOC1 and THOC2 bound less RNA from short intronless or short-first-exon pachytene, pre-pachytene, or hybrid piRNA precursor transcripts (1.6 and 1.4 times more than protein-coding transcripts, respectively, Wilcoxon rank-sum test $p$ value $< 1.1 \times 10^{-13}$). Remarkably, for all seven spliced piRNA clusters with long first exons, the RIP signals of

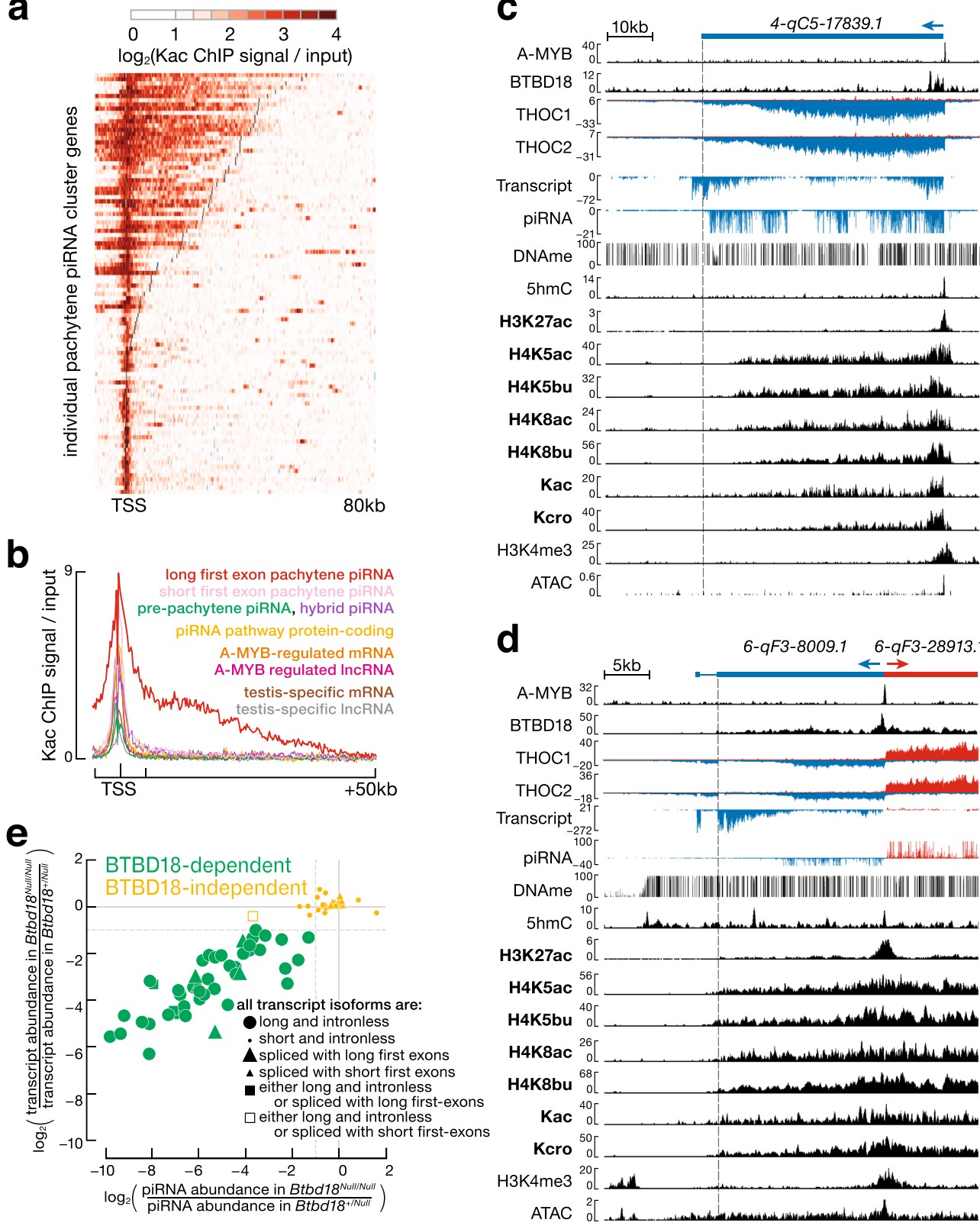

THOC1 and THOC2 were largely confined to long first exons of the piRNA precursor transcripts, suggesting that the THO complex binds to transcripts from the 5′-end to the first 5′ splice site (Fig. 4b and Supplementary Fig. 6a).

As described in the previous section, the pachytene piRNA gene *10-qB4-6488.1* and the protein-coding gene *Ddx50* are divergently transcribed from a common A-MYB- and BTBD18-bound

promoter. The first exon of *10-qB4-6488.1* is 50,297 bp long, whereas the first exon of *Ddx50* spans just 187 bp. Supporting the hypothesis that the THO complex marks transcripts for entry into the piRNA biogenesis pathway, THOC1 and THOC2 bound across the *10-qB4-6488.1* first exon, but did not detectably bind the *Ddx50* mRNA (Supplementary Fig. 5d). Another pachytene piRNA gene described in the previous section, *pi-1700016M24Rik.1*, produces

**Fig. 3 Half of pachytene piRNA clusters have long first exons, or are long and intronless and display high histone acylation signals across their first exon or gene body. a** Heatmap reports the enrichment of the pan-lysine acetylation ChIP signal relative to input in pachytene spermatocytes for 100 pachytene piRNA clusters in the −10 kb to +80 kb window flanking the TSS. Black bar, first exon–intron boundary; blue bar, 3′ end of transcript. **b** A meta-gene plot shows average lysine acetylation signal in pachytene spermatocytes for long-first-exon (or long intronless) pachytene piRNA clusters, short-first-exon pachytene piRNA clusters, and other groups of genes in the −10 kb to +80 kb window flanking the TSS. **c** UCSC genome browser view of *4-qC5-17839.1*, a long, intronless pachytene piRNA-producing gene with a low-CG promoter. All signal tracks are from pachytene spermatocytes except for BTBD18, THOC1, THOC2, and 5hmC, which are from adult testis. **d** UCSC genome browser view of *6-qF3-8009.1*, an intron-containing, long first exon, piRNA-producing gene with a low-CG promoter. Vertical dashed line marks the end of the first exon. **e** Scatter plot compares the change in transcript and piRNA abundance for *Btbd18* mutant compared to *Btbd18* heterozygous testis. Green: BTBD18-dependent pachytene piRNA clusters; yellow: BTBD18-independent pachytene piRNA clusters. The transcripts of pachytene piRNA clusters are further classified as long and intronless ($n = 39$); short and intronless ($n = 13$); spliced with long first exons ($n = 5$); spliced with short first exons ($n = 40$); intronless transcripts isoforms are long and spliced transcript isoforms have long first exons ($n = 2$); intronless transcript isoforms are long and spliced transcript isoforms have short first exons ($n = 1$).

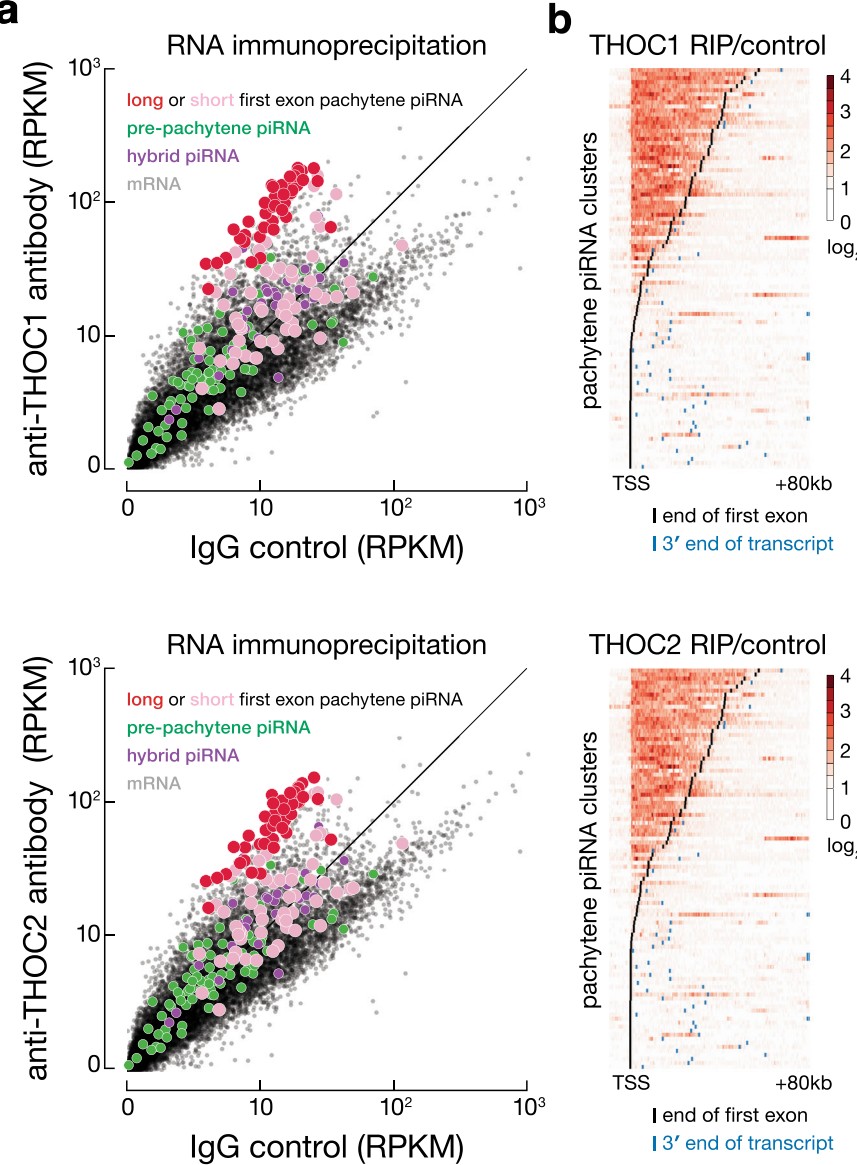

**Fig. 4 THOC1 and THOC2 specifically bind precursor transcripts from long intronless and long-first-exon pachytene piRNA clusters. a** RIP-seq was used to measure the abundance of RNA bound to THOC1 ($n = 5$) or THOC2 ($n = 3$) compared to IgG control ($n = 4$) in adult testis. Each dot represents the mean abundance normalized for transcript length (RPKM). **b** Heatmaps report the enrichment of THOC1 or THOC2 binding, measured by RIP-seq, relative to IgG control for pachytene piRNA clusters in adult testis. TSS, transcription start site. Each row corresponds to pachytene piRNA gene, in first-exon (spliced) or transcript (unspliced) length order. For each gene, the black bar denotes the first exon–intron boundary; the blue bar indicates the 3′ end of the transcript.

three RNA isoforms: a 10,179 bp unspliced transcript that appears to be the source of piRNAs and two spliced mRNAs with short first exons. Normalized for transcript length, THOC1 and THOC2 bound six times more to the long intronless transcript than either of the short-first-exon transcripts (Supplementary Fig. 5c).

**In the testis, low-CG pachytene piRNA clusters have high levels of histone acylation and 5-hydroxymethylcytosine.** Nearly half (49/100) of pachytene piRNA clusters have low-CG promoters (Fig. 2c); these genes are moderately enriched in the long intronless or long-first-exon group (30/49; Chi-square test $p$ value = $8.8 \times 10^{-3}$) and BTBD18 dependence (31/49; Chi-square test $p$ value = $4.8 \times 10^{-3}$). Unlike protein-coding genes with low-CG promoters, the low-CG pachytene piRNA clusters maintain high histone acylation levels in adult testis. They also exhibit high 5hmC levels, which may aid their transcriptional activation[49].

Consistent with the known correlation between the expression level of a gene and the O/E CG of its promoter, protein-coding genes with low-CG promoters had significantly lower steady-state mRNA abundance in the testis than those with intermediate-CG or high-CG promoters (Supplementary Fig. 6b). Accordingly, these low-CG promoters have higher levels of DNA methylation, lower levels of active histone marks, reduced chromatin accessibility, and less RNA pol II binding than intermediate- or high-CG promoters (Supplementary Fig. 6b). Pachytene piRNA clusters defy the rule that promoters with low O/E CG and high DNA methylation are poorly expressed in the testis; instead, the pachytene piRNA clusters with low-CG, highly methylated promoters were as highly expressed and made as many piRNAs as those loci with intermediate- or high-CG promoters (Fig. 5a). Furthermore, low-CG pachytene piRNA clusters had significantly higher levels of histone acylation, despite significantly lower levels of H3K4me3, chromatin accessibility, and RNA pol II binding, compared with loci having intermediate- or high-CG promoters (Fig. 5b). These paradoxical results support the view that, in the testis, histone acylation overrides other determinants of transcription, allowing the expression of pachytene piRNA clusters irrespective of the O/E CG or methylation level of their promoters.

How do low-CG pachytene piRNA clusters maintain high levels of histone acylation? In the testis, highly expressed genes with low promoter O/E CG, but high levels of H3K4me3 in their promoters also have high levels of 5hmC[21], a DNA modification leading to transcriptional activation[49–51]. Supporting this idea, pachytene piRNA clusters with low-CG promoters had high 5hmC levels. Using published hMeDIP data which specifically measure 5hmC levels (see "Methods"), we found that the promoter 5hmC levels of low-CG pachytene piRNA clusters were significantly higher than those of control genes (i.e., low-CG genes that do not produce piRNAs) in cells where pachytene piRNAs are abundant—pachytene spermatocytes (Wilcoxon rank-sum test $p$ value < $9.7 \times 10^{-7}$) and round spermatids ($p$ value < $5.0 \times 10^{-6}$)—but not in hepatocytes, which lack piRNAs (Fig. 5c and Supplementary Fig. 6c, d). We conclude that despite their hypermethylation, pachytene piRNA clusters with low-CG promoters can be expressed to high levels in the testis because of compensating 5hmC DNA modification.

**Genic and epigenomic features correlated with silencing of pachytene piRNA clusters in the soma.** What prevents expression of pachytene piRNA clusters in somatic cells, which generally lack the machinery to process precursor transcripts into piRNAs? We found that two genic features associated with pachytene piRNA clusters—low-CG promoters and long first exons—correlated with their somatic silencing. Furthermore, we identified two epigenomic signals—DNA methylation and the repressive histone mark H3K27me3—likely underpin this silencing.

The promoters of both pachytene piRNA clusters and testis-specific protein-coding genes were highly methylated (>50%) in the neonatal forebrain, heart, and liver—representative somatic tissues. Yet a subset of these two classes of genes showed lower levels of promoter methylation in pachytene spermatocytes, representative of male germline cells (Fig. 5d). These subsets of genes with different methylation levels between germline and soma mostly had promoters with intermediate levels (0.25–0.5) of O/E CG (Supplementary Fig. 7). Specifically, the median methylation level for the pachytene piRNA clusters with intermediate-CG promoters was 16% in pachytene spermatocytes vs. 72–81% in somatic tissues (Fig. 5d; Wilcoxon rank-sum $p$ values < $3.2 \times 10^{-6}$). Similarly, the median methylation level for the testis-specific protein-coding genes with intermediate-CG promoters was 17% in pachytene spermatocytes vs. 71–81% in the soma ($p$ values < $2.2 \times 10^{-16}$). These results suggest that DNA methylation may be an essential mechanism for silencing these intermediate-CG genes in the soma.

Indeed, most low-CG promoters of pachytene piRNA clusters (40/49) were highly methylated in both pachytene spermatocytes and forebrain, and most high-CG promoters (17/18) showed low levels of methylation in both pachytene spermatocytes and the forebrain. In contrast, most intermediate-CG (23/33) and some low-CG (9/49) promoters had low methylation in pachytene spermatocytes, but high methylation in the forebrain (Fig. 5e and Supplementary Fig. 7). Consistent with the idea that genes with highly methylated, low-CG promoters are silenced in the soma, the steady-state forebrain transcription levels were <0.1 RPKM for all of the 77 pachytene piRNA clusters with high DNA methylation in the forebrain. Further supporting this idea, all of the 12 pachytene piRNA clusters with forebrain transcript abundance ≥0.1 RPKM had intermediate- or high-CG promoters with low levels of DNA methylation. Our analysis in two other somatic tissues, neonatal heart, and liver, revealed the same repressive effect of DNA methylation (Supplementary Fig. 8a). Thus, DNA methylation likely silences a majority of pachytene piRNA clusters in the soma, especially those with low- or intermediate-CG promoters.

Among the 23 pachytene piRNA clusters with low DNA methylation in the forebrain, 11 showed some degree of testis-specific expression—their forebrain transcript abundance was <0.1 RPKM. All of these 11 genes have high ($n = 8$) or intermediate- ($n = 3$) CG promoters. For these genes, H3K27me3 rather than DNA methylation may repress somatic expression: the median H3K27me3 level on the promoters of these genes was 1.9 times higher in the forebrain than in pachytene spermatocytes (Wilcoxon rank-sum $p$ value = $9.8 \times 10^{-4}$). For the remaining 12 pachytene piRNA clusters expressed in forebrain, which constituted all of the pachytene piRNA clusters with forebrain transcript abundance ≥0.1 RPKM, promoter H3K27me3 levels in forebrain and pachytene spermatocytes do not significantly differ (Fig. 5f). Similar results were observed in neonatal heart and liver (Supplementary Fig. 8b). Thus, high H3K27me3 may be another mechanism for silencing pachytene piRNA clusters in the soma, especially those with high-CG and intermediate-CG promoters that are not DNA methylated in the soma.

The presence of long first exons may be yet another mechanism for somatic silencing of pachytene piRNA clusters with high-CG and intermediate-CG promoters lacking somatic DNA methylation. The 11 genes in this subset that are silenced in the forebrain have substantially longer first exons than the 12

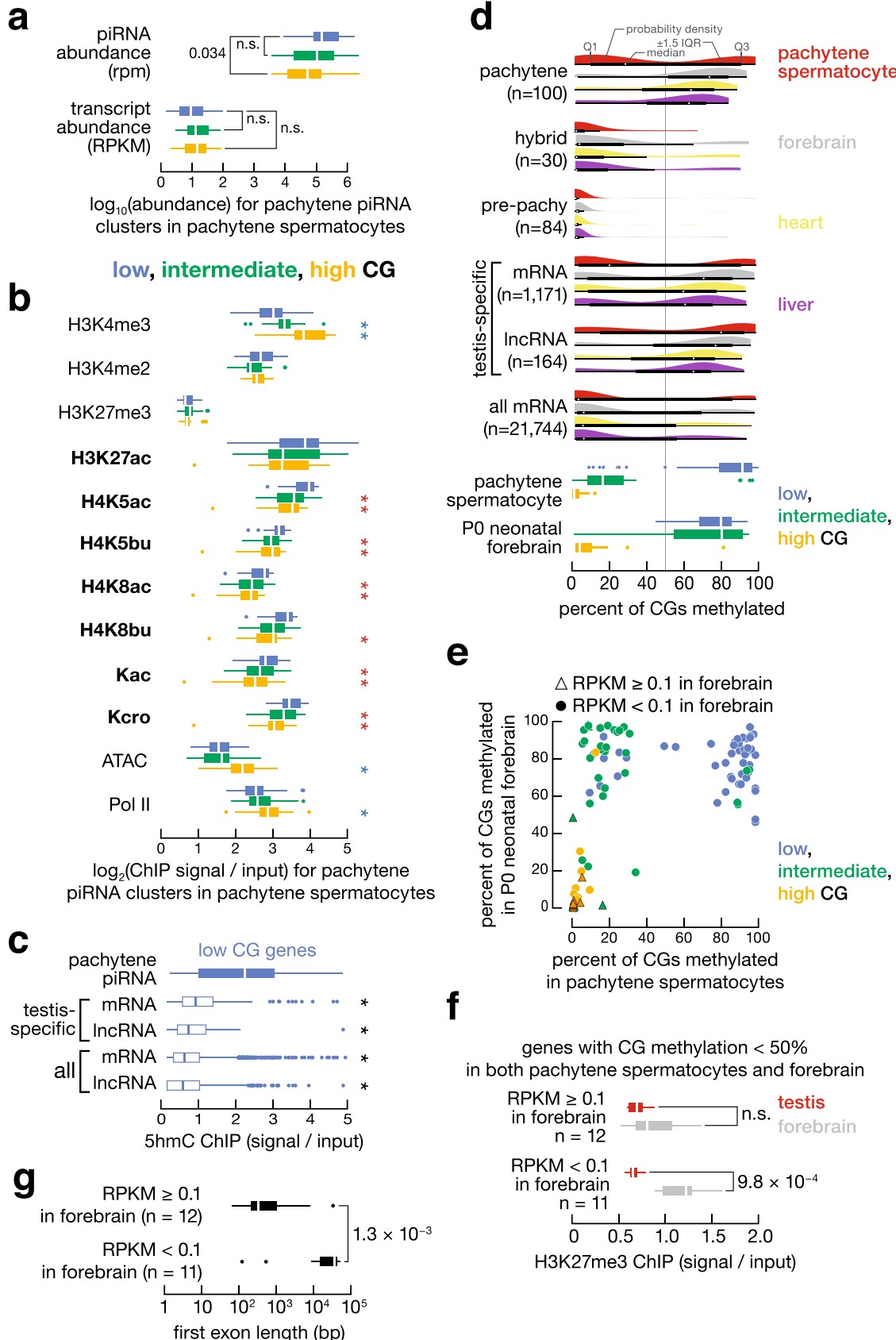

genes that are expressed in the forebrain (Fig. 5g; median first-exon length = 35,227 vs. 308; Wilcoxon rank-sum test $p$ value = $1.3 \times 10^{-3}$). Similar results were observed in neonatal heart and liver (Supplementary Fig. 8c).

In theory, the absence of A-MYB in somatic cells could explain why pachytene piRNA clusters are not transcribed outside the germline. However, most non-pachytene piRNA cluster loci that are regulated by A-MYB in testis ($n = 930$, defined in "Methods")

**Fig. 5 Activation of pachytene piRNA clusters in the male germline correlates with high 5hmC levels, whereas the extent of somatic suppression of these genes correlates with first-exon length and high levels of DNA methylation and H3K27me3. a** piRNA and transcript abundance for pachytene piRNA clusters in pachytene spermatocytes. Promoter O/E CG is indicate by color: Blue, low ($n = 49$); green, intermediate ($n = 33$); yellow, high ($n = 18$). Two-sided Wilcoxon rank-sum test was used to determine $p$ value; n.s. not significant. For all boxplots in this figure, whiskers show 95% confidence intervals, boxes represent the first and third quartiles, and the vertical midline is the median. **b** Level of histone modifications, ATAC, and RNA pol II for pachytene piRNA clusters in pachytene spermatocytes. Promoter O/E CG is as in **a**. Asterisks indicate two-sided Wilcoxon rank-sum test $p$ values < 0.05 between 49 low-CG and 33 intermediate- or 18 high-CG promoters. Red (blue) indicates that the epigenetic level is significantly higher (lower) in low-CG pachytene piRNA clusters. **c** 5hmC levels in pachytene spermatocytes. Asterisks indicate two-sided Wilcoxon rank-sum test $p$ values < 0.05 between 49 pachytene piRNA clusters and 397 testis-specific mRNAs, 95 testis-specific lncRNAs, all 6177 mRNAs, and all 2050 lncRNAs with low-CG promoters. **d** Top, DNA methylation levels for pachytene piRNA clusters and other groups of genes. Bottom, DNA methylation levels of pachytene piRNA clusters grouped as in **a**. Vertical line indicates the 50% DNA methylation level. **e** DNA methylation levels of pachytene piRNA clusters between forebrain and pachytene spermatocytes. Colors indicate promoter CG level; genes with forebrain expression level higher than 0.1 RPKM are denoted as triangles. **f** H3K27me3 levels, relative to control, are shown in adult testis and forebrain. Only the 23 pachytene piRNA clusters in the bottom-left quadrant defined in **e** are included and further classified by their expression levels in the forebrain: RPKM ≥ 0.1 ($n = 12$) or RPKM < 0.1 ($n = 11$). **g** First exon length of the same 23 pachytene piRNA clusters as in **f**, classified into the same groups as in **f**.

are expressed in somatic tissues (696 of 930 with >1 RPKM), despite low levels of *A-Myb* transcript in somatic cells (0.54 RPKM). The *A-Myb* paralogs *B-Myb* and *C-Myb* bind similar sequence motifs to *A-Myb*, but they are expressed in only a subset of the somatic tissues (*B-Myb* in colon and spleen and *C-Myb* in colon, lung, and spleen among the tissues we examined); therefore, they are unlikely to be responsible for the expression of A-MYB target genes in somatic tissues. We conclude that the absence of A-MYB in the soma is not sufficient to silence pachytene piRNA clusters in somatic tissues. Thus, three mechanisms—low CG and the associated high hypermethylation, high H3K27me3, and long first exons—may explain the somatic silencing of pachytene piRNA clusters. We performed the same analysis for testis-specific protein-coding genes, and observed that high DNA methylation and high H3K27me3, but not long first exons (Fig. 5d and Supplementary Fig. 9), likely repress these genes in the soma: among 1171 testis-specific protein-coding genes, 452 and 157 were expressed in the forebrain at >0.1 and >1 RPKM, respectively. Thus, testis-specific protein-coding genes and pachytene piRNA-producing genes may be silenced in the soma by similar mechanisms.

**Evolutionarily conserved features of pachytene piRNA clusters**. Although individual piRNA sequences are rarely conserved across species, the genomic location (synteny), promoter elements, and exon–intron structure of the major pachytene clusters are preserved among placental mammals[12]. Notably, long, unspliced and long-first-exon mouse pachytene piRNA clusters are more likely to be syntenic in other species than those with short first exons. We did not detect significant conservation of low-CG promoters between the species examined.

We analyzed the 89 pachytene piRNA clusters in human, 133 in rhesus, 132 in marmoset, 114 in rat, 117 in cow, 194 in opossum, and 89 in platypus[12] for their evolutionary conservation with the 100 mouse pachytene piRNA clusters. We determined whether the genes were syntenic and whether the syntenic genes also produced piRNAs. For 21 mouse pachytene piRNA clusters, the syntenic loci in at least three other eutherian species produced piRNAs at similar levels (<5-fold change from mouse). For 45 other mouse pachytene piRNA clusters, the syntenic loci in rat produced similar levels of piRNAs. These data define three groups of pachytene piRNA clusters: eutherian-conserved, murine-conserved, and mouse-specific (34 pachytene piRNA clusters; Fig. 6a). Among the 100 mouse pachytene piRNA clusters, most eutherian-conserved loci had long first exons, while most murine-conserved and mouse-specific loci had short first exons (Fig. 6a; Fisher's exact test $p$ value = $6.2 \times 10^{-5}$). In each of the six eutherian mammals examined, eutherian-conserved pachytene

piRNA clusters had longer first exons than other pachytene piRNA clusters (Fig. 6b; Wilcoxon rank-sum test $p$ values < $1.3 \times 10^{-3}$), suggesting that a long first exon promotes pachytene piRNA production across eutherian mammals.

The 47 long intronless or long-first-exon mouse pachytene piRNA clusters show high levels of histone acylation extending across their first exon or gene body. High acylation is evolutionarily conserved among pachytene piRNA-producing loci in *Rhesus macaque*, which diverged from mouse 90 million years ago. H3K27ac ChIP-seq in adult Rhesus testis revealed that H3K27 acetylation levels were higher at pachytene piRNA-producing genes than other gene types, and the H3K27ac signal spread broadly from the TSS as far as 40 kbp (Fig. 6c). Moreover, a long, unspliced transcript and shared synteny correlated with high levels of piRNA production across placental mammals. Just 12–18% of the pachytene piRNA clusters in each of the six eutherian mammals have long first exons and are related by synteny with pachytene piRNA-producing loci in at least three of the six other species examined; yet this small set of genes produce 45–76% pachytene piRNAs in adult testis (Fig. 6d).

## Discussion

Pachytene piRNA clusters, which are unique to placental mammals, are expressed almost exclusively in the male germline. Of the 100 mouse pachytene piRNA clusters, 47 are long and unspliced or have long first exons, 49 have low-CG promoters, and 30 display both features. In general, a lack of splicing or a low O/E CG promoter is anticipated to prevent somatic expression, and indeed, genes with these features are silenced in the soma. Yet pachytene piRNA clusters with these same features are highly expressed in the germline, and produce most of the pachytene piRNAs in the adult mouse and other eutherian mammals.

Low-CG promoters are hypermethylated and recognized by proteins with methyl-CG-binding domains (MBDs). MBD proteins, in turn, recruit histone deacetylases (HDACs), especially class III HDACs, such as sirtuins, which remove lysine acetyl, butyryl, and crotonyl modifications[52]. Decreased histone acylation correlates with transcriptional repression[53,54]. Although the promoters of low-CG pachytene piRNA clusters are hypermethylated in the testis, they are also highly 5-hydroxymethylated, a modification that can block binding of MDB proteins. Consequently, 5-hydroxymethylation prevents binding of HDACs[50]. We suggest that the high 5-hydroxymethylation of the low-CG promoters of pachytene piRNA clusters allows them to maintain high lysine acylation and active transcription in the male germline.

Three additional features likely act to repress pachytene piRNA clusters in the soma. First, both low-CG and intermediate-CG promoters show high levels of DNA methylation in the soma.

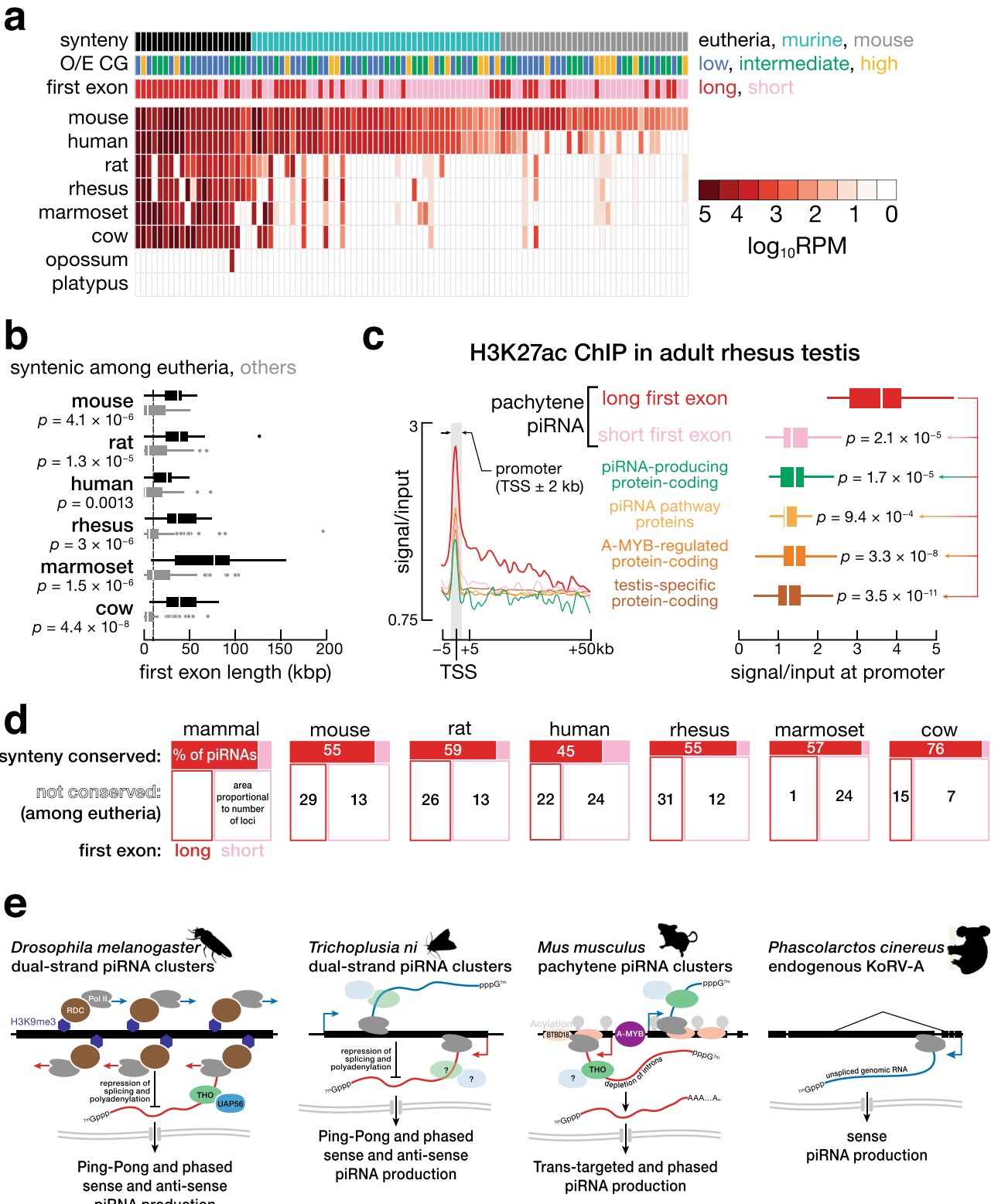

Second, the subset of pachytene piRNA clusters with high- or immediate-CG promoters show a high density of the repressive histone mark H3K27me3. Third, pachytene piRNA clusters tend to have long first exons or long unspliced transcripts. Pachytene piRNA clusters with intermediate-CG promoters may particularly rely on DNA methylation and H3K27me3 for their somatic silencing: just one-third (11/33) of intermediate-CG pachytene piRNA clusters are unspliced with long transcripts or spliced with

long first exons; nearly two-thirds (30/49) of low-CG pachytene piRNA clusters display these features.

Most mammalian promoters inherently drive bidirectional transcription. Co-transcriptional binding of the spliceosomal U1 snRNP enhances transcriptional elongation in the productive direction[55,56]. Long first exons delay the recruitment of the splicing machinery and hence hinder transcriptional elongation. Co-transcriptional splicing has been proposed to be required for

**Fig. 6 Long first exons and broad histone acylation domains are conserved features of pachytene piRNA clusters in placental mammals. a** Heatmap shows piRNA abundance (RPM) for the 100 mouse pachytene piRNA clusters and the syntenic loci in several mammalian species with increasing evolutionary distance. Pachytene piRNA clusters are ranked first by conservation and then by average piRNA abundance. **b** First exon length for pachytene piRNA clusters in mouse, rat, human, rhesus, marmoset, and cow. Two-sided Wilcoxon rank-sum *p* values are provided for comparing eutherian-conserved with other pachytene piRNA clusters in mouse ($n = 21$ and $79$), rat ($n = 20$ and $94$), human ($n = 19$ and $70$), rhesus ($n = 18$ and $115$), marmoset ($n = 17$ and $115$), and cow ($n = 18$ and $99$). Vertical line marks 10 kbp. **c** Left, meta-gene plot shows average H3K27ac signal in rhesus adult testis across the $-5$ to $+50$ kb window flanking the TSS for six groups of genes: 54 long-first-exon pachytene piRNA clusters, 79 short-first-exon pachytene piRNA clusters, 56 piRNA clusters that overlap protein-coding genes, 17 piRNA pathway genes, 593 A-MYB-regulated protein-coding genes, and 745 testis-specific protein-coding genes. Right, boxplots show the H3K27ac density relative to input for the same six groups of genes. **d** Mosaic plots for pachytene piRNA clusters. Filled boxes, eutherian-conserved pachytene piRNA clusters; unfilled boxes, other pachytene piRNA clusters. Red, long-first-exon and long intronless pachytene piRNA clusters; pink, short-first-exon and short intronless pachytene piRNA clusters. The area of each patch indicates the number of pachytene piRNA clusters in the group; numbers report the percentage of total piRNAs in each group. **e** Schematic diagram shows our proposed model that suppression of splicing steers piRNA precursor transcripts into the piRNA biogenesis pathway across animal species. Credit: silhouettes in **e** are from http://phylopic.org. The cabbage looper silhouette was by Gareth Monger (https://creativecommons.org/licenses/by/3.0/). The silhouettes have not been altered in any way. For all boxplots, whiskers show 95% confidence intervals, boxes represent the first and third quartiles, and the vertical midline is the median. Two-sided Wilcoxon rank-sum tests were used to calculate *p* values.

the recruitment of the THO complex, which is essential for transcriptional elongation and nuclear export of spliced RNA[57]. Consistent with this model, deletion of the first intron of a protein-coding gene markedly decreases its expression[58]. How then do long unspliced and long-first-exon pachytene piRNA transcripts recruit the THO complex in the male germline? Our analyses identify several features that may promote THO complex binding in the absence of splicing: broad histone acylation and binding of the transcriptional elongation factor BTBD18. Perhaps BTBD18 recruits histone acyl transferases to open the chromatin of long-first-exon or unspliced pachytene piRNA clusters. We do not yet know whether the efficient transcriptional elongation that results from BTBD18 binding and histone acylation suffices to allow co-transcriptional loading of the THO complex or if BTBD18 also participates in the direct recruitment of THO. Once bound to the piRNA precursor transcript, the THO complex may enhance transcriptional elongation and facilitate export of the piRNA precursor transcripts to the perinuclear nuage, where piRNA processing initiates (Fig. 6e). Clearly a major question for future study is how BTBD18 or other proteins identify pachytene piRNA precursor transcripts.

Suppression of splicing is a unifying feature of piRNA-producing loci across the animal kingdom. In flies, transcription of the heterochromatic piRNA clusters relies on a noncanonical mechanism, in which binding of the protein Rhino to the chromatin mark H3K9me3 allows promoter-independent recruitment of RNA pol II[59–62]. The complex of proteins assembled around Rhino also serves to suppress splicing and to bypass cleavage at polyadenylation signal sequences[61–64]. Lepidoptera lack this noncanonical transcription machinery, and data from the *Trichoplusia ni* germ cell line Hi5 suggest that piRNA clusters are transcribed from standard RNA pol II promoters, yet the piRNA precursors from these clusters also undergo little splicing[65]. In koala males bearing a KoRV-A provirus, piRNAs are produced only from the unspliced proviral transcript[66]. Similarly, the long-first-exon or long unspliced pachytene piRNA precursor transcripts of mice and other placental mammals are inherently depleted of introns (Fig. 6e). The link between piRNA production and suppression of splicing calls to mind the antagonism between splicing and siRNA generation in *Cryptococcus neoformans*, where the two pathways compete for RNA substrates[67]. We suggest that a failure to engage the splicing machinery near the transcription start marks transcripts for processing into piRNAs. Because most protein-coding genes contain short first exons and are efficiently spliced, such a mechanism would protect mRNAs from inappropriately entering the piRNA pathway. Moreover, channeling unspliced transcripts into the

piRNA pathway would allow the unspliced genomic transcripts of retroviruses and retrotransposons to serve as precursors for sense piRNAs. Conversion of the unspliced transcripts to sense piRNAs—which cannot target the viral or transposon transcripts directly via sequence complementarity—would serve to destroy the RNA transcript, preventing accumulation of viral or transposon genomic RNA that can catalyze retro-insertion into the host genome[66]. Such a mechanism would provide an initial defense against these pathogens until the establishment of an antisense piRNA response that can directly target the spliced, protein-coding transcripts of retroviruses and transposons.

## Methods

**Mice**. All mice were maintained and used according to the guidelines of the Institutional Animal Care and Use Committee of the University of Massachusetts Medical School. C57BL/6 J mice (RRID: IMSR_JAX:000664) were used as wild-type controls.

**H3K27ac ChIP-seq**. To perform ChIP, small pieces of frozen testis tissue were cross-linked with 2% (w/v) formaldehyde at room temperature for 30 min using end-over-end tumbler. We next crushed fixed tissues in the presence of ChIP lysis buffer (1% (w/v) SDS, 10 mM EDTA, 50 mM Tris-HCl, pH 8.1) by 40 strokes with a B pestle in a Dounce homogenizer (Kimble-Chase, Vineland, USA). Thereafter, lysate was sonicated using Covaris ultrasonicator (Covaris, E220) to shear the chromatin to 150–200 bp. Lysate was then diluted 1:10 with ChIP dilution buffer (0.01% (w/v) SDS, 1.1% (w/v) Triton X-100, 1.2 mM EDTA, 16.7 mM Tris-HCl, pH 8.1, 167 mM NaCl). We performed immunoprecipitation using 6 μg rabbit polyclonal anti-H3K27Ac antibody (Abcam, ab4729). Following the immunoprecipitation, we extracted DNA with phenol:chloroform:isoamyl alcohol (25:24:1; pH 8) and prepared libraries for anti-H3K27ac, and input DNA as previously described[68]. ChIP-seq libraries were sequenced as 79-nt paired-end reads using NextSeq500 (Illumina).

**THOC1 and THOC2 RIP-seq**. Testis from 4-month-old C57BL/6 J mice were frozen in liquid nitrogen and stored at $-80\,°C$ until use. One testis was lysed in 1 ml lysis buffer (50 mM HEPES-KOH, pH 7.5, 150 mM KCl, 3.2 mM MgCl$_2$, 0.5% (v/v) NP-40, 1 mM phenylmethylsulphonyl fluoride; 1× Proteinase Inhibitor cOmplete, EDTA-free Protease Inhibitor Cocktail (Roche), 0.4 U/μl RNAseOUT (Invitrogen)) in 10 strokes with pestle A and 20 strokes with pestle B in a 2 ml Dounce homogenizer (Kimble-Chase). Lysate was sonicated (Bioruptor, Diagenode) at medium strength, 15 s on and 1 min off for 10 min at $0\,°C$. Lysate was cleared by centrifugation for 15 min at $16,050 \times g$ at $4\,°C$. Lysate (100 μl) was used for input RNA extraction using RNeasy Mini Kit (Qiagen). A total of 10 μl rabbit anti-THOC2 and anti-IgG, or mouse anti-THOC1 and anti-IgG antibody was conjugated to the 100 μl Dynabeads (Invitrogen) protein A or protein G, respectively, in citric phosphate (CP) buffer (7.10 g Na$_2$HPO$_4$, 11.5 g citric acid in 1 l water, pH 5.6) for 2 h at room temperature with rotation. Antibody-conjugated beads were washed three times with CP buffer containing 0.1% (v/v) Tween-20 and incubated with mouse testis lysate overnight at $4\,°C$ with rotation. Beads were washed three times in cold ($4\,°C$) lysis buffer. The immunoprecipitated RNA was recovered from the antibody beads using the RNeasy Mini kit (Qiagen). The extracted RNA was used to generated RNA-seq libraries as described[68].

**Data processing**. We reanalyzed previously published RNA-seq[31,69], small RNA-seq[70], whole-genome bisulfite sequencing or WGBS[21], ChIP-seq of histone marks[42,45,71], ChIP-seq of RNA pol II (ref. [72]), and ATAC-seq[73] data. We also reanalyzed ovary RNA-seq data (E11.5, E12.5, E14.5, E16.5, E18.5, and E20.5) from GEO (accession: GSE119411). Public ChIP-seq data of H3K4me1, H3K4me3, H3K27me3, H3K27ac, and RNA pol II in six adult mouse tissues (testis, liver, heart, kidney, lung, and spleen) were obtained from the mouseENCODE consortium[74]. We also included H3K4me2 ChIP-seq data for the adult mouse testis, liver, and brain[69], but we did not find ChIP-seq data in other somatic tissues to fully complement the RNA-seq data. We analyzed the WGBS data generated by the ENCODE consortium on fetal mouse tissues at eight embryonic time points until birth (E10.5 to P0), along with published WGBS data on mouse round spermatids[21].

**Gene annotation**. Gene annotations and ribosomal RNA (rRNA) sequences were from Ensembl (Release 82). To determine whether Ensembl Release 100 might yield more lncRNAs with long first exons, we reanalyzed the data using the latest Ensembl gene annotation. Indeed, Ensembl Release 100 contains 924 more lncRNAs than Release 82 (5,586 vs. 4,662), but just one of the 924 lncRNAs (AC160336.1) has long first exon. RNA-seq data indicate that this lncRNA is not expressed in adult mouse testis. Thus, our conclusions remain unchanged using Ensembl Release 100.

**RNA-seq and RIP-seq data**. To analyze RNA-seq and RIP-seq data, rRNA was first removed from RNA-seq and RIP-seq data by mapping to rRNAs using Bowtie2 (version 2.2.5) with default parameters[75]. Unmapped reads were then mapped to the mm10 genome from the UCSC genome browser using STAR (version 020201) with default parameters (allowing up to 2 mismatch and up to 100 mapping locations)[76]. SAMtools (version 1.8) was used to transform the alignment results from sam format to bam format throughout this study[77]. HTSeq (version 0.9.1) was used with default parameters to count uniquely mapping reads[78]. For each protein-coding or lncRNA gene (Ensembl Release 82) and piRNA-producing locus, we then calculated and normalized the number of uniquely mapped reads as Reads Per Million mapped reads (RPM) and the number of uniquely mapped reads normalized by the total transcript length of each gene as RPKM.

**ChIP-seq and ATAC-seq data**. ChIP-seq data and ATAC-seq data were mapped to the mouse reference genome mm10 using Bowtie2 with the parameter–very-sensitive. Enrichment of ChIP signal over input (.bedGraph) was computed from alignment files (.bam) across the mouse genome using the MACS2 callpeak and bdgcmp modules with default parameters[79]. The H3K27ac ChIP-seq data in primary spermatocytes and round spermatids (accession GSE107398) were not provided with input, so we calculated read density normalized by sequencing depth. Furthermore, we normalized read density by sequencing depth for all ATAC-seq data. To calculate the level of each epigenetic mark for the various groups of genes (piRNA clusters, protein-coding genes, lncRNA genes, and their subsets), we averaged the enrichment or read density at the promoter (TSS ± 2 kb) or gene body (TSS + 2 kb till the 3′-end) as indicated.

We used a control gene set to normalize ChIP-seq signal levels across different datasets. We first identified 1230 genes with similar expression levels (0.66 < change < 1.5 and maximal expression >0.1 RPKM) in three cell types (spermatogonia, primary spermatocytes, and round spermatids). We then used the median levels of each histone mark for these 1230 control genes in three cell types to normalize the levels of all histone marks, chromatin accessibility (ATAC signal), and RNA pol II binding in the corresponding cell types. We used the same strategy to normalize the H3K27me3 levels in testis and somatic tissues. We identified a set of control genes whose expression levels differed by <1.5-fold between testis and each somatic tissue: 1132, 913, and 817 control genes for comparing testis with forebrain, heart, and liver, respectively. This normalization method reduced the bias caused by the difference in antibody specificity across different batches or different tissue or cell types.

**WGBS data**. We applied quality control and adapter trimming to all WGBS data using Trim Galore! (version 0.4.1). Sequence alignment and methylation calls were performed using Bismark (version v0.15.0, no mismatch allowed; mapping statistics in Supplementary Data 3) with default parameters[80]. Only C in CG positions was used for computing methylation levels. Methylation level was computed as mCG/CG averaged over all detected CG sites in a region. Each WGBS dataset had four replicates, and we calculated the methylation level for each region in each replicate and then averaged the levels across the four replicates.

**Small RNA-seq data**. Small RNA sequencing data were mapped to the mouse genome (mm10) using Bowtie (version 1.1.0) after removing rRNA (from Ensembl) and miRNA (from miRBase) with parameters -a–best–strata[81]. Only 24–32 nt small RNAs were retained. We then calculated the piRNA abundance of each piRNA gene using BEDTools intersectBed, apportioning the reads that

mapped to multiple locations[82]. piRNA abundance was reported normalized to the total number of reads mapping uniquely to one genomic location. As most of the mapped reads are uniquely mapped to the mouse genome (90.6% in adult testis, 89.7% in pachytene spermatocyte, and 90.0% in round spermatid), normalizing to uniquely mapping or uniquely mapping plus multiply mapping reads result in similar piRNA abundance and the same conclusions.

**5hMe-DIP data**. 5hMe-DIP (5hmC) data were mapped to the mouse genome (mm10) using Bowtie2 with the parameter–very-sensitive. The number of aligned reads were processed into enrichment relative to input in the bedGraph format. The 5hmC level for each gene was calculated by averaging the enrichment in the TSS ± 500 bp window.

**Definition of genes regulated by A-MYB**. A-MYB-regulated genes were defined based on A-MYB ChIP-seq data on wild-type mice, and long RNA-seq data on *A-Myb* mutant and heterozygous mice. We first identified the genes whose TSSs were within 500 bp of a A-MYB ChIP-seq peak called by MACS2 with parameters -q 0.05–keep-dup all -B (ref. [79]). We then filtered out the genes with less than threefold enrichment of A-MYB ChIP-seq signal over input in the TSS ± 500 bp window. Keeping only those genes with >2-fold decrease in *A-Myb* mutant testis compared to heterozygotes at both 14.5 and 17.5 dpp yielded 837 A-MYB-regulated genes, including 791 protein-coding and 46 lncRNA genes (Supplementary Data 2).

**Definition of high-, intermediate- and low-CG promoter classes**. Normalized CG dinucleotide content in a promoter was calculated as described previously[83]: the ratio of observed to the expected number of CG dinucleotides in the promoter (O/E CG), where the expected number = [(Fraction of C + Fraction of G)/2]², i.e.,

$$\mathrm{O/E\,CG} = \frac{\text{Fraction of CpG}}{\left[\left(\text{Fraction of C} + \text{Fraction of G}\right)/2\right]^2}. \tag{1}$$

Promoters were classified into three groups: low-CG, O/E CG < 0.25; intermediate-CG, 0.25 < O/E CG < 0.5; and high-CG, O/E CG > 0.5.

**Definition of testis-specific genes**. Testis-specific genes corresponded to those genes with expression levels in testis >4-fold higher than the maximal expression in any somatic tissues that we examined. We also calculated a tissue-specific score (ts-score),

$$\mathrm{ts-score} = \frac{\sum_{i=1}^{n}\left(1 - \frac{\mathrm{Exp}_i}{\mathrm{Exp}_{\max}}\right)}{n - 1}, \tag{2}$$

where $n$ denotes the total number of tissues examined, $\mathrm{Exp}_i$ denotes the expression levels in a particular tissue, and $\mathrm{Exp}_{ts}$ denotes the expression level in testis (we set the ts-score to 0 if any $\mathrm{Exp}_i > \mathrm{Exp}_{ts}$). We used ts-score to rank piRNAs in Fig. 1a.

**Refining the annotations of piRNA clusters**. Based on the RNA-seq and small RNA-seq data produced after 2013, we made small modifications to our previous annotations[11] of several piRNA clusters (Supplementary Data 4): (1) we removed the intron in *12-qE-23911.1* because it is in a low mappability region and not supported by uniquely mapping reads. (2) We removed the intron in *5-qF-14224.1* which is 59 nt long with a noncanonical splice site motif CC-CT and supported by few reads. (3) We changed the TSS of *17-qC-59.1* from chr17: 50,237,659 to chr17:50,239,160 based on A-MYB ChIP-seq, RNA-seq, and small RNA-seq data. (4) We changed the main TSS of *4-qB3-639.1* from chr4: 62,230,936 to chr4: 62,228,511 based on A-MYB ChIP-seq, RNA-seq, small RNA-seq, and BTBD18 ChIP-seq data. (5) We added a long and intronless isoform (chr2: 92,529,805–92,540,950) for *Gm13817*, which is divergently transcribed from *2-qE1-35981.1*. Gm13817 is an unannotated pachytene piRNA-producing gene and produces >100 piRNAs per million unique mapped reads. (6) We removed the long-first-exon isoform of *pi-Zfp652.1* (a hybrid piRNA clusters), which is not supported by RNA-seq or small RNA-seq reads. (7) We changed the main TSS of *10-qC-875.1* from chr10: 86,617,011 to chr10: 86,591,510 based on A-MYB and H3K4me3 ChIP-seq, RNA-seq, small RNA-seq, and BTBD18 ChIP-seq data. We also separated *10-qC-875.1* from a spliced lncRNA *Gm48485*, which is primarily expressed in round spermatid.

**Definition of BTBD18-dependent pachytene piRNA clusters**. We defined BTBD18-dependent pachytene piRNA clusters as those whose expression level was ≥2-fold lower in pachytene spermatocytes of *Btbd18* mutant mice than *Btbd18* heterozygous mice, and whose piRNA abundance was also ≥2-fold lower in the 18-day postpartum testis tissue of *Btbd18* mutant mice than that of *Btbd18* heterozygous mice.

**Annotation of piRNA clusters in human, rhesus, marmoset, rat, cow, rabbit, opossum, and platypus**. Human piRNA clusters were annotated previously[12]. We refined the previously annotated piRNA clusters in rhesus, marmoset, rat, cow,

opossum, and platypus[12] to delineate exons and introns (Supplementary Data 4). We considered 24–32 nt small RNA reads that could map to each mammalian genome, after rRNA, miRNA, tRNA, snRNA, and snoRNA were removed, as piRNAs. piRNA abundance was then computed for 20 kb sliding windows (with a 1 kb step) in the genome, and windows with >100 piRNAs per million uniquely mapped piRNAs were deemed potential piRNA clusters. To remove false positives due to unannotated miRNA, rRNA, tRNA, snRNA, or snoRNA, which mostly produce reads with the same sequences, we also filtered out the 20-kb genomic windows with fewer than 200 distinct reads. We then calculated the first-nucleotide composition for each 20-kb window and discarded those windows with fewer than 50% of its piRNAs having a 1 U or 10 A (with the 10 A possibly resulting from ping-pong amplification). The remaining contiguous 20-kb windows were deemed putative piRNA clusters. To obtain the precise promoter position of each piRNA gene, we performed trimming from the 5′ and 3′ ends by examining adjacent 100-bp windows. The first and last two nearby windows (closer than 1000 bp) each with more than two piRNAs per million uniquely mapped piRNAs were considered as the 5′ and 3′ ends of the piRNA gene. We used the RNA-seq reads after rRNA removal for annotating introns of piRNA clusters. Finally, we performed manual curation for each piRNA gene using piRNA profile, long RNA profile, and exon–exon junctions detected using RNA-seq reads. The transcriptional direction of a piRNA gene was indicated by the direction of the main long RNA transcript. The final piRNA clusters were classified according to their genomic location, and those with >50% base pairs overlapping protein-coding genes were defined as genic and the rest as intergenic. We treated intergenic piRNA clusters as pachytene piRNA clusters in rhesus, marmoset, rat, cow, opossum, and platypus. We defined 17 piRNA pathway genes in rhesus, 747 testis-specific protein-coding genes, 593 A-MYB-regulated protein-coding genes using one-to-one orthology between mouse and rhesus.

**Evolutionary conservation of pachytene piRNA clusters**. We performed evolutionary conservation analysis on mouse pachytene piRNA clusters with a similar approach as described before for human piRNA clusters[12]. Briefly, we mapped the 100 mouse pachytene piRNA clusters to human, rhesus, marmoset, rat, cow, opossum, and platypus genomes, using UCSC chain files and liftOver with the parameter -minMatch = 0.1 and recorded the syntenic regions of pachytene piRNA clusters that could be lifted over to each of the other species[12]. The coordinates of the syntenic region in the other species which overlapped a mouse piRNA-producing gene on the same genomic strand were adjusted to be the piRNA gene coordinates we annotated in that species. To be inclusive of piRNAs that map to the boundaries of the syntenic regions in that other species that did not overlap piRNA clusters on the same genomic strand, we extended these syntenic regions by 10 kb in both ends and then calculated the piRNA abundance in the regions, using small RNA-seq data in that other species. Finally, eutherian-conserved pachytene piRNA clusters were defined as those for which three or more eutherian mammals (among human, rhesus, marmoset, rat, or cow) could produce similar amounts of piRNAs (change < 5) to mouse in the syntenic regions. The remaining pachytene piRNA clusters were defined as murine-conserved when the syntenic region in rat produced similar amount of piRNAs to mouse, mouse-specific if otherwise.

**Calculation of extension index and relative extension index**. In order to quantify how much the signals of histone modification, chromatin accessibility, or factor binding extend into the gene body, we first cut the genome into 200-bp bins and then computed the average enrichment of each type of signal over input or average read count per one million reads over these 200-bp bins. Bins far downstream pachytene piRNA clusters (TSS + 80 kb to TSS + 100 kb) were considered background bins and bins with signal higher than 95% quantile of the background bins were regarded as signal-enriched bins. Transposon copies reside in pachytene piRNA clusters can hinder the signal identification and lead to signal gaps. To avoid this, we identified the furthest continually enriched bins for each pachytene piRNA clusters allowing 3800-bp gaps (19 bins). The extension index was defined as the distance from the TSS to the furthest continually enriched bin. The relative extension index was defined as the extension index divided by the relative first-exon length, with the maximum set to 1.

**Reporting summary**. Further information on research design is available in the Nature Research Reporting Summary linked to this article.

## Data availability

The THOC2 RIP-seq data in adult mouse testis and the H3K27ac ChIP-seq data in adult rhesus testis have been deposited in the GEO with the accession GSE147724. All public data used in this study were downloaded from GEO and ENCODE, and are shown in Supplementary Fig. 10 and Supplementary Data 3 with their accessions. The data supporting the findings of this study are available from the corresponding authors upon reasonable request.

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

## Acknowledgements

We thank the members of the Weng, Zamore, and Theurkauf laboratories for their critical comments. This work was supported in part by Chinese National Natural Science Foundation grants (31571362 and 31871296) to Z.W. and National Institutes of Health grant P01 HD078253 to W.E.T, Z.W., and P.D.Z.

## Author contributions

T.Y., K.F., P.D.Z., and Z.W. conceived the project. T.Y. and K.F. performed computational analyses. Y.F. aided computational analyses. D.M.O. performed the *Rhesus* H3K27ac ChIP-seq experiment. G.Z. and W.E.T. performed the THOC2 RIP-seq experiment. T.Y., K.F., P.D.Z., and Z.W. wrote the manuscript.

## Competing interests

The authors declare no competing interests.
