## [Peer Review File · Nature Communications]

REVIEWER COMMENTS

Reviewer #1 (Remarks to the Author):

In trying to answer the central questions of “what marks pachytene piRNAs genes to produce piRNAs, and what confines their expression to the germline”, this study conducts an integrative analysis of transcriptome, piRNA and histone modification marks and DNA methylation patterns, along with THO complex pulldowns and re-analysis of BTBD18 and A-Myb mutants in mostly mouse and few other mammalian species to search for a set of defining features for pachytene piRNA clusters. As the authors state, they found long first exons or unspliced long transcripts, certain histone acetylation marks, and a more methylated DNA promoter status as CORRELATED with pachytene piRNA production. I capitalize ‘Correlate’ since this study frequently extends these ‘Correlations’ to imply causation with statements like in the title “Steer” and in the abstract “these features explain why...”.

This study does present a high-quality, rich compendium of data as a useful resource of transcriptomic and epigenetic features around mammalian piRNA clusters, including new datasets like THO complex RNA IP datasets and many histone modifications marks and CpG methylation patterns in pachytene spermatocytes. The figures are well done, and the manuscript is well written with the correlation findings described clearly. Although the BTBD18 and A-Myb transcription factor roles in piRNA transcription stimulation and THO complex binding to piRNA precursor transcripts have already been described by other previous studies including by some of these authors, the new data in this study is somewhat descriptive or correlative. Although the correlations are very intriguing, there is no other experimental data to test the paper’s claim of splicing inhibition or histone acetylation as truly causative signals of piRNA generation.

I am cognizant of recent lab closures during the pandemic may have halted many experiments, although some labs are starting to reopen. However, I would have wanted to see stronger, less descriptive analyses bolster the transcriptomic/epigenetic features as drivers of piRNA generation. A definitive but probably very demanding experiment would be to genetically manipulate a transcript locus that is highly expressed in the mouse testes but does not make piRNAs to transform it into a poorly-splicing and more extensively acetylated locus to see if that also converts it into making piRNAs.

Alternatively perhaps, the authors could look at more extensive transcriptomic and epigenetic data from other mammals where the orthologous loci are clearly present between mouse and the other mammals, but then focus on the species-specific differentially-expressed piRNA clusters to see if the transcriptomic/epigenetic features also track with the specific specie. Would the low CG and high hypermethylation, high H3K27me3 and shorter exons also track with differentially in species-specific piRNA cluster differences and explain when a given transcript is now silenced from entering the piRNA pathway?

Overall, I think this study provides a nice additional characterization of epigenetic marks around pachytene piRNA loci in mice, but it requires at least a revision that can provide some stronger

experimental evidence to bolster the correlative claims.

Reviewer #2 (Remarks to the Author):

Dear editor,

In this manuscript entitled “Long First Exons and Epigenetic Marks Steer Transcripts Into the Pachytene piRNA Pathway” Yu, Fan and colleagues studied a set of ~100 pachytene piRNA cluster genes in mouse. They have compared exon length, expression patterns, Pol II occupancy, histone marks, transcription factor binding, and DNA methylation (5mC) and hydroxymethylation (5hmC) across these pachytene piRNA cluster genes to other types of genes. Furthermore, they shown that cluster genes with a long first exon are most different from other genes.

As expected, the pachytene piRNA cluster genes are expressed mainly in testis where they show active histone marks (Fig 1, 2), while being decorated with suppressive marks in other tissues. A higher level of histone marks associated with active transcription is correlated to a higher piRNA output (Fig 2). The most interesting observations include that mainly piRNA cluster genes with a long first exon are BTBD18 dependent. Furthermore, the THO complex binding is stronger for this group, and this subset of piRNA cluster genes shows a higher conservation across other species (Fig 3, 4, 6).

The study is mainly computational, but details on how the analysis was performed are largely missing. For instance, this reviewer could not find any mention on how differentially expressed genes were identified nor how binding sites were derived from some of the ChIP-seq data. Those aspects are key to the interpretation of the rest of the manuscript. Moreover, while other analyses are described in terms of which tool that was used (such as TopHat2, Trim Galore!, MACS2, bdgcmp, Bismark, ...), the parameters are not specified and no scripts are provided with the manuscript.

Many results are shown as genome browser screenshots, average profiles and correlations. Those correlation and differences between groups of genes are often interpreted as being causal (e.g., in the abstract “Together, these features explain why the major sources of pachytene piRNA loci specifically generate these unique small RNAs in the male germline of placental mammals.”). The causality is rarely proven by the data and analyses. Furthermore, differences in average signal may be driven by a small subset of loci and this makes it difficult to evaluate how generalizable the results were across the piRNA clusters. It would also be useful to show how different features co-occur. Several mutually exclusive features were all found as enriched at piRNA cluster genes (H3K4 me1, me2 and me3; H4K4 and K5 ac and bu; 5mC and 5hmC). This needs to be discussed.

Many published datasets were used and the supplementary file should include individual accession numbers for each dataset. Currently only study accessions are provided and some data are missing entirely (e.g., H3K4me1). Additional details to show that the data is of sufficient quality and that the

replicates are consistent would strengthen the analysis.

Overall, while being mostly descriptive, the findings reported in this work should be of interest to the readership of Nature Communications. However, I strongly suggest that the following comments and questions are addressed before this manuscript is accepted for publication.

Specific comments/questions

- 1) The authors should revise their nomenclature as the terms “piRNA genes” and “piRNA pathway genes” are confusing. Instead of using “piRNA genes” the term “piRNA source loci” should be used to clearly distinguish it from the genes coding for piRNA pathway factors.
- 2) Page 6, line 145: “Conversely, pachytene piRNA genes had 1.3–2.0-fold lower levels of the repressive histone mark H3K27me3 in testis than in somatic tissues” - This signal looks like noise. Is the difference driven by a single site?
- 3) This reviewer found “CG content” a confusing way to refer to the observed vs expected ratio of CpG dinucleotides at a given promoter. Please use the established CpG nomenclature to refer to the dinucleotides and make it clear when you describe the observed/expected CpG ratio. Similarly, it seems like terms such as “high CG” is currently used to refer to high methylation levels at CpG sites. Please make it clear throughout the manuscript what is being referred to.
- 4) How relevant is the CpG methylation in promoters with very few sites?
- 5) The formula for the expected number of CpG dinucleotides ($[(C\%+G\%)/2]^2$) seems to be off by a factor 100. This reviewer also cannot understand why the C and G frequencies were averaged. A much simpler $C \times G$ formula would be more accurate.
- 6) Fig 2A & C: Use of tables in figure panels should be avoided, perhaps the authors could instead summarize the displayed statistics in a heatmap?
- 7) Fig 2B: The figure is difficult to interpret since most genes fall into a single bar. Change x-axis in log₁₀-scale or find another way to illustrate the length distributions.
- 8) Fig 3A: The panel is lacking the colour legend on top.
- 9) Fig 3C & D and Fig S4: all genome browser screenshots are lacking y-axis. These need to be added for each track.
- 10) The authors used an old gene annotation to analyse the exon lengths (Ensembl 82 from October 2015) – would using a more recent annotation find more lncRNAs with long first exons?
- 11) Was the 5hMe-DIP data normalized to the number of CpG sites?
- 12) Page 14, line 359: The claim that “for spliced transcripts, THOC1 and THOC2 binding was largely confined to long first exons of the piRNA precursor” needs to be substantiated. This is not clear from Fig 4B since the location of the following exons are not clearly indicated. Mapping to the transcriptome or quantification per exon would be required.
- 13) The piRNA abundance normalization could be problematic. According to the description, multi-mapping reads were kept for the analysis, but they were excluded during normalization. This should be clarified.
- 14) The RNA-seq and RIP-seq alignment is described as “allowing up to 2 mismatch and up to 100 mapping locations”. It is generally not advisable to count the same read multiple times. The authors

should clarify the mapping parameters. How did this affect the downstream analysis and interpretation?

Minor comments

Are the number of piRNA cluster loci described in the introduction correct? For instance, the number of pre-pachytene clusters reported in Özata et al (2020) was much higher than 83.

- 1) "79 paired-end reads" should be "79 nt paired-end reads"
- 2) "Trim Galore" should be "Trim Galore!"
- 3) "and 30 display both features" should be "and 3 display both features"
- 4) It would be useful to include a supplementary table listing the refined piRNA loci in mouse and other species.
- 5) What does "Note: Mapped data were downloaded directly from the ENCODE portal." in Table S3 refer to?

Reviewer #3 (Remarks to the Author):

Yu et al. present a systematic evaluation of the sequence features and epigenetic landscapes that distinguish pachytene piRNA precursor gene expression, and targeting of these genes for piRNA production, in adult male testes. It has previously been determined that a small number of piRNA precursor genes are the source of piRNA production in germline cells. However, many questions have remained about how the expression of pachytene piRNA precursors is specified to the male germline, how the cell identifies certain genes for piRNA production, and what keeps these pathways silent in somatic tissues. This study combines re-analysis of a large number of public datasets as well as three newly generated RIP- and ChIP-seq datasets to try to determine whether certain genomic and epigenomic features distinguish piRNA precursors from other germline expressed transcripts (and silencing of these loci in the soma).

Curious features that appear to distinguish pachytene piRNA loci include: high levels of DNA methylation despite relatively low CG dinucleotide content, high levels of 5 hydroxymethylation in addition to CG methylation, abundantly expressed transcripts from genes that are either unspliced or contain very long first exons, and high levels of histone acylation that appears to be broadly distributed around the promoters and across the gene bodies of these regions. It is still unclear which of the epigenetic marks is laid down first, which epigenetic writers lay down these marks, and what proteins recognize these features as piRNA specific. However, this provides some very nice new information about the distinguishing features of piRNA precursors that produce the most abundant piRNAs.

I do have some questions that I think need to be addressed in order to make the story more clear.

- 1) In Figure 1A, in the paragraph starting on Line 218, and in several places through the text, much is

made of the high methylation levels of pachytene piRNA gene promoters. Yet, there is very little graphical display of CG methylation levels across these genes. Instead, we get a correlation coefficient and p-value table in Figure 2A (btw, hopefully, multiple hypothesis corrections have been made to this series of correlation p-values – Bonferroni or BH?). Since this is a major conclusion, it seems like this data merits display, preferably in wiggle/profile format. I did see the CG methylation graphs in Fig 5D-E for pachytene spermatocytes and a few somatic tissues. But there seem to be several other tissues with WGBS data in Supplemental table 8, and the discussion of 5mC levels begins long before Fig 5 is introduced. More specifically, since these precursors have the rather uncommon feature of showing both high 5hmC and high 5mC, is there a way of displaying relative levels of the two types of methylation on piRNA precursor promoters versus other genes? Or in the germline versus soma?

2) Beginning on line 256, there's discussion of the "broadness" of several epigenetic marks, extending far into the piRNA gene bodies. Yet, if you look at Fig 2D, these marks don't seem predominantly asymmetric with a bias in the direction of transcription. They simply seem to cover a broad region on both sides of the TSS. Moreover, in several of the CHIP heatmaps and wiggle plots, these are cut off a few kb upstream of the TSS, but clearly extend in both directions. Is this primarily due to genes with bidirectional promoters that are driving the upstream enrichments? Or, is this broad enrichment more common across even unidirectionally transcribed piRNA genes? I guess I'm asking if the boundaries of the broad histone patterns correlate with RNA/transcription boundaries? Or if instead, the epigenetic landscapes that define piRNA gene loci occupy a larger surrounding region and potentially influence (or are restricted by) surrounding genes?

3) For Fig 4B, I was struck by the wording that describes THOC1/2 binding that seems to end at the first splice site or 3' end (whichever is first). You've put it down as THOC1/2 being loaded before intron removal but after recognition of the 5' splice site. Is the data not also consistent with a scenario that says THOC1/2 could be loaded anytime until the polymerase reaches the first 5' splice site or 3' processing site? Perhaps there's additional arguments or literature on THO complex function that guided your choice of words, but I'm curious about the rationale for this interpretation.

4) Given the refinements to the piRNA gene annotations described in the methods, which sound very reasonable, I think it would be a nice thing for the field if an updated table of piRNA gene structures was given as an additional supplemental table. Ideally, it would be nice if this were in bed, GTF, or GFF format.

REVIEWER COMMENTS

Reviewer #1 (Remarks to the Author):

1.1 In trying to answer the central questions of “what marks pachytene piRNAs genes to produce piRNAs, and what confines their expression to the germline”, this study conducts an integrative analysis of transcriptome, piRNA and histone modification marks and DNA methylation patterns, along with THO complex pulldowns and re-analysis of BTBD18 and A-Myb mutants in mostly mouse and few other mammalian species to search for a set of defining features for pachytene piRNA clusters. As the authors state, they found long first exons or unspliced long transcripts, certain histone acetylation marks, and a more methylated DNA promoter status as CORRELATED with pachytene piRNA production. I capitalize ‘Correlate’ since this study frequently extends these ‘Correlations’ to imply causation with statements like in the title “Steer” and in the abstract “these features explain why...”.

This study does present a high-quality, rich compendium of data as a useful resource of transcriptomic and epigenetic features around mammalian piRNA clusters, including new datasets like THO complex RNA IP datasets and many histone modifications marks and CpG methylation patterns in pachytene spermatocytes. The figures are well done, and the manuscript is well written with the correlation findings described clearly. Although the BTBD18 and A-Myb transcription factor roles in piRNA transcription stimulation and THO complex binding to piRNA precursor transcripts have already been described by other previous studies including by some of these authors, the new data in this study is somewhat descriptive or correlative. Although the correlations are very intriguing, there is no other experimental data to test the paper’s claim of splicing inhibition or histone acetylation as truly causative signals of piRNA generation.

I am cognizant of recent lab closures during the pandemic may have halted many experiments, although some labs are starting to reopen. However, I would have wanted to see stronger, less descriptive analyses bolster the transcriptomic/epigenetic features as drivers of piRNA generation. A definitive but probably very demanding experiment would be to genetically manipulate a transcript locus that is highly expressed in the mouse testes but does not make piRNAs to transform it into a poorly-splicing and more extensively acetylated locus to see if that also converts it into making piRNAs.

Response:

Indeed, the results of our computational analysis are correlational and do not prove causation. We have toned down our claim in the paper by removing “steer” from the title and adding “may” before “explain” in the last sentence of the abstract. Now the title reads: “Long First Exons and Epigenetic Marks Distinguish Pachytene piRNA clusters from Other Mammalian Genes”.

Thank you for your thoughtful suggestion of an experiment—indeed, creating a transgenic mouse by altering a locus for its splicing and acetylation could be a powerful experimental test of our model. As you have also noted, such an endeavor would be very time consuming, especially under the current COVID-19 pandemic, but we plan to start such experiments and report the results in the future.

1.2 Alternatively perhaps, the authors could look at more extensive transcriptomic and epigenetic data from other mammals where the orthologous loci are clearly present between mouse and the other mammals, but then focus on the species-specific differentially-expressed piRNA clusters to see if the transcriptomic/epigenetic features also track with the specific species. Would the low CG and high hypermethylation, high H3K27me3 and shorter exons also track with differentially in species-specific piRNA cluster differences and explain when a given transcript is now silenced from entering the piRNA pathway?

Overall, I think this study provides a nice additional characterization of epigenetic marks around pachytene piRNA loci in mice, but it requires at least a revision that can provide some stronger experimental evidence to bolster the correlative claims.

Response:

Thank you for your suggestion. Human and mouse have the most complete transcriptomic and epigenetic data as well as manually curated gene structures of piRNA clusters (exons, introns, transcriptional starts, and transcriptional ends), but epigenetic data are mostly unavailable and gene structures are imprecise for piRNA clusters in other mammals. Lack of precise definition of transcription starts hinders accurate evaluation of CG and DNA methylation levels at promoters. Nevertheless, in our original submission, we analyzed first exon lengths in all six eutherian mammals and found it to be a conserved feature for pachytene piRNA clusters (Fig. 6b), and we further produced H3K27ac ChIP-seq data in rhesus testis and detected enrichment of this histone modification at pachytene piRNA clusters with long first exons (Fig. 6c).

Motivated by your comment, we have now carefully examined the synteny of each piRNA cluster between human and mouse in search of species-specific piRNA cluster differences, and the results are detailed in Reviewer's Table 1 (appended to the end of this file and also uploaded as an Excel file). Specifically, we looked for piRNA clusters whose first exons were long and transcribed in one species (human or mouse) but short and transcribed at the syntenic location in the other species. Only one piRNA cluster—*7-qD1-9417.1* in mice—shows such species-specific differences. It makes long (35,373 nt), unspliced transcripts in mouse but generates a spliced transcript with a short (424 nt) first exon in humans. Reviewer's Figure 1 (next page) shows that *7-qD1-9417.1* makes abundant piRNAs in mice (10,816 ppm) but few piRNAs in humans (148 ppm), supporting our model of long first exons being correlated with RNAs entering the piRNA pathway. We also attempted to define the gene structures of the syntenic loci in other mammals. The syntenic locus in rat makes a long, unspliced transcript that produces abundant piRNAs (9,587 ppm), while the syntenic loci in rhesus and marmoset make short-first-exon transcripts and less abundant piRNAs (~2,200 ppm). The syntenic locus in cow is unspliced, although it is difficult to determine precisely the transcript length; it makes slightly more piRNAs than rhesus and marmoset (2,777 ppm). We could add these data to the revised manuscript if you think they are valuable, but we have decided to leave them out for now because a single locus does not provide statistically significant support and we must await experimental results in the future.

Reviewer's Figure 1. Steady state transcript and piRNA profiles of a pachytene piRNA cluster 7-*qD1-9417.1* in mouse and syntenic regions of this piRNA cluster in rat, human, rhesus, marmoset, and cow. Red indicates that the piRNA cluster is on the plus genomic strand and blue indicates minus genomic strand. Note that the orientations in humans and Rhesus are reversed.

Reviewer #2 (Remarks to the Author):

In this manuscript entitled “Long First Exons and Epigenetic Marks Steer Transcripts Into the Pachytene piRNA Pathway” Yu, Fan and colleagues studied a set of ~100 pachytene piRNA cluster genes in mouse. They have compared exon length, expression patterns, Pol II occupancy, histone marks, transcription factor binding, and DNA methylation (5mC) and hydroxymethylation (5hmC) across these pachytene piRNA cluster genes to other types of genes. Furthermore, they shown that cluster genes with a long first exon are most different from other genes.

As expected, the pachytene piRNA cluster genes are expressed mainly in testis where they show active histone marks (Fig 1, 2), while being decorated with suppressive marks in other tissues. A higher level of histone marks associated with active transcription is correlated to a higher piRNA output (Fig 2). The most interesting observations include that mainly piRNA cluster genes with a long first exon are BTBD18 dependent. Furthermore, the THO complex binding is stronger for this group, and this subset of piRNA cluster genes shows a higher conservation across other species (Fig 3, 4, 6).

2.1 The study is mainly computational, but details on how the analysis was performed are largely missing. For instance, this reviewer could not find any mention on how differentially expressed genes were identified nor how binding sites were derived from some of the ChIP-seq data. Those aspects are key to the interpretation of the rest of the manuscript. Moreover, while other analyses are described in terms of which tool that was used (such as TopHat2, Trim Galore!, MACS2, bdgcmp, Bismark, ...), the parameters are not specified and no scripts are provided with the manuscript.

Response:

The Methods section now describes the detailed parameters used for each tool (STAR, Trim Galore!, MACS2, Bismark, Bowtie and Bowtie2). For example, we identified ChIP-seq peaks using MACS2 with parameters -q 0.05 --keep-dup all -B.

2.2 Many results are shown as genome browser screenshots, average profiles and correlations. Those correlation and differences between groups of genes are often interpreted as being causal (e.g., in the abstract “Together, these features explain why the major sources of pachytene piRNA loci specifically generate these unique small RNAs in the male germline of placental mammals.”). The causality is rarely proven by the data and analyses. Furthermore, differences in average signal may be driven by a small subset of loci and this makes it difficult to evaluate how generalizable the results were across the piRNA clusters. It would also be useful to show how different features co-occur. Several mutually exclusive features were all found as enriched at piRNA cluster genes (H3K4 me1, me2 and me3; H4K4 and K5 ac and bu; 5mC and 5hmC). This needs to be discussed.

Response:

We have toned down our statement that can be interpreted as causation. Now the title reads, “Long First Exons and Epigenetic Marks Distinguish Pachytene piRNA Clusters from Other Mammalian Genes”. The last sentence of the Introduction reads, “Together, these features may explain why the major sources of pachytene piRNA loci specifically generate these unique small RNAs in the male germline of placental mammals.” We have carefully examined

the entire manuscript and made sure that we use “correlated” and similar phrases throughout.

We plotted average profiles and correlations to save space. We agree that sometimes average profiles and correlations can be misleading due to a small number of outliers; however, this is not the case for our results and most pachytene piRNA clusters behave like the average profile. We have added to Supplementary Fig. 1 heatmap that shows the H3K27ac signal at individual loci, corresponding to the average H3K27ac profile in Fig. 1b. As you can see from this heatmap, the difference in average signals between the germline and the soma is not driving by a small subset of pachytene piRNA clusters but is evident across all pachytene piRNA clusters. Additionally, we have added statistical significance for comparing piRNA clusters when average profiles and correlation coefficients are shown (Fig. 1b caption, Fig. 2a and Supplementary Fig. 4c).

Fig. 3c, d and Supplementary Fig. 5 show the co-occurrence of histone acylations and BTBD18 across the first exon of several long-first-exon pachytene piRNA clusters. These figures also show co-occurrence of promoter marks such as ATAC, H3K4me3 and Pol II around the transcription start sites at these pachytene piRNA clusters.

You are right; several mutually exclusive histone marks were found as enriched at piRNA clusters (H3K4 me1, me2 and me3; H4K4 and K5 ac and bu; 5mC and 5hmC), and we now state in Results that these histone marks cannot exist on the same lysine residue of the same nucleosome but are rather at the piRNA clusters in different subsets of germ cells.

Supplementary Fig. 1.
H3K27ac profiles at individual pachytene piRNA clusters.

A heatmap shows the enrichment of the H3K27ac ChIP signal relative to input in adult testis, heart, kidney, liver, and spleen for 100 pachytene piRNA clusters in the -2 kb to +2 kb window flanking the TSS. Each pachytene piRNA cluster is represented by one row of the heatmap, in the same order across the tissues.

2.3 Many published datasets were used and the supplementary file should include individual accession numbers for each dataset. Currently only study accessions are provided and some data are missing entirely (e.g., H3K4me1). Additional details to show that the data is of sufficient quality and that the replicates are consistent would strengthen the analysis.

Response:

We have now added accession for each dataset in a new Supplementary Table 3, along with mapping statistics for each dataset showing that the data are of sufficient quality.

2.4 Overall, while being mostly descriptive, the findings reported in this work should be of interest to the readership of Nature Communications. However, I strongly suggest that the following comments and questions are addressed before this manuscript is accepted for publication.

Response:

Thank you for your positive comment! We have made every effort to address your comments and questions.

2.5 The authors should revise their nomenclature as the terms “piRNA genes” and “piRNA pathway genes” are confusing. Instead of using “piRNA genes” the term “piRNA source loci” should be used to clearly distinguish it from the genes coding for piRNA pathway factors.

Response:

We have now changed “piRNA genes” to “piRNA clusters”, which is the widely used term for “piRNA source loci” in the piRNA field. Line 62 now reads: “The genomic regions that produce piRNAs, referred to as piRNA clusters or piRNA source loci, ...” We originally used “piRNA genes” to emphasize that these piRNA clusters are well annotated in mice with their transcription start sites, poly-A sites, and intron positions. However, we do agree with you that “piRNA genes” sound too similar to “piRNA pathway genes” and may cause confusion.

2.6 Page 6, line 145: “Conversely, pachytene piRNA genes had 1.3–2.0-fold lower levels of the repressive histone mark H3K27me3 in testis than in somatic tissues” - This signal looks like noise. Is the difference driven by a single site?

Response:

The difference, although small in magnitude, was not driven by a single or a few piRNA clusters. We performed a Wilcoxon signed-rank test which compared all piRNA clusters between testis and each of the somatic tissues, and the differences are significant (p-values ranged from 2.3×10^{-11} in kidney to 7.9×10^{-6} in spleen).

2.7 This reviewer found “CG content” a confusing way to refer to the observed vs expected ratio of CpG dinucleotides at a given promoter. Please use the established CpG nomenclature to refer to the dinucleotides and make it clear when you describe the observed/expected CpG ratio. Similarly, it seems like terms such as “high CG” is currently used to refer to high methylation levels at CpG sites. Please make it clear throughout the manuscript what is being referred to.

Response:

Indeed, in the previous submission, by CG content, we meant the observed/expected CpG ratio. We have now changed “CG content” to “O/E CG” throughout the manuscript to make the definition clearer. The term “high CG” means “high CpG content” previously, or “high O/E CG” after the renaming. We never intended to use “high CG” to mean high methylation levels at CpG sites; in contrast, “high CG” is typically correlated with low levels of DNA methylation.

2.8 How relevant is the CpG methylation in promoters with very few sites?

Response:

There are in total 6,164 genes (including mRNAs, lncRNAs, piRNA clusters, snoRNAs, snRNAs, etc.) that possess promoters with 1–5 CpGs. The methylation levels of these promoters are ~100% (Reviewer’s Figure 2 below).

2.9 The formula for the expected number of CpG dinucleotides (“ $[(C\%+G\%)/2]^2$ ”) seems to be off by a factor 100. This reviewer also cannot understand why the C and G frequencies were averaged. A much simpler $C \cdot G$ formula would be more accurate.

Response:

Indeed, the percent sign (“%”) in this equation is not correct; it should have been fraction. We have changed the equation to:

$$\frac{\text{Fraction of CpG}}{[(\text{Fraction of C} + \text{Fraction of G})/2]^2}$$

We used the formula of expected fraction of CpG dinucleotides as defined in a previous publication (Landolin J.M et al., *Genome Res*, 2010). The reason that the fraction of C and the fraction of G are averaged is to account for the double-stranded nature of genomic DNA. Because the reverse complement of the CpG dinucleotide is also CpG, the Fraction of CpG in a region would be the same regardless whether one uses the reference genomic sequence or its reverse complement sequence for the calculation; thus, the expected fraction also should not be dependent upon which genomic strand was used for performing the calculation. When the reference genomic sequence in a region has more G's than C's, the reverse complement genomic sequence would have more C's than G's, and the average removes this unbalance between the two genomic strands in computing the expected fraction of CpG dinucleotides. Now this is clearly explained in Methods.

2.10 Fig 2A & C: Use of tables in figure panels should be avoided, perhaps the authors could instead summarize the displayed statistics in a heatmap?

Response:

Thank you for the suggestion. Instead of tables, we've changed Fig. 2A & C to barplots.

2.11 Fig 2B: The figure is difficult to interpret since most genes fall into a single bar. Change x-axis in log₁₀-scale or find another way to illustrate the length distributions.

Response:

We've changed the x-axis of Fig 2B into log₁₀ scale.

2.12 Fig 3A: The panel is lacking the colour legend on top.

Response:

You are right. We've added a color legend at the top of Fig 3A.

2.13 Fig 3C & D and Fig S5: all genome browser screenshots are lacking y-axis. These need to be added for each track.

Response:

We've added the y-axis for each track in genome browser screenshots in Fig 3C & D and Fig S5.

2.14 The authors used an old gene annotation to analyse the exon lengths (Ensembl 82 from October 2015) – would using a more recent annotation find more lncRNAs with long first exons?

Response:

The latest Ensembl gene annotation (Release 100) has 924 more lncRNAs than version 82 (5,586 vs 4,662). Compared with Release 82, the latest Release 100 only yielded one more lncRNA (AC160336.1) with a long first exon. However, RNA-seq data in adult mouse testis indicate that the lncRNA (AC160336.1) is not expressed in adult testis. Thus, our previous

conclusion stays the same with the latest Ensembl gene annotation. We have added this information in Methods.

2.15 Was the 5hMe-DIP data normalized to the number of CpG sites?

Response:

No, we did not normalize 5hMe-DIP to the number of CpG sites. Promoters with more CpGs do tend to have higher 5hMe-DIP signals. However, the numbers of CpG dinucleotides in the promoters of five groups of genes—low-CG pachytene piRNA clusters, low-CG testis-specific protein-coding genes, lncRNAs, all low-CG protein-coding genes, and lncRNAs—are similar. A figure of 5hMe-DIP signals normalized by the number of CpGs (Reviewer’s Figure 3 below) looks identical to the previous Fig. 5c and Supplementary Fig. 6d, which were not normalized by CpG count.

related to Figure 5c

related to Supplementary Figure 5b

Reviewer’s Figure 3. 5hmC levels of pachytene piRNA genes normalized by the number of CG dinucleotides in the promoters of testis-specific mRNAs and lncRNAs and all mRNAs and lncRNAs in pachytene spermatocytes, round spermatids, sperm and liver.

2.16 Page 14, line 359: The claim that “for spliced transcripts, THOC1 and THOC2 binding was largely confined to long first exons of the piRNA precursor” needs to be substantiated. This is not clear from Fig 4B since the location of the following exons are not clearly indicated. Mapping to the transcriptome or quantification per exon would be required.

Response:

We’ve quantified THOC1 and THOC2 binding in the first exon and other exons from long-first-exon pachytene piRNA clusters (7 loci in total). A new figure (Supplementary Fig. 6a in the revised manuscript, also shown below) shows THOC1 and THOC2 binding is restricted to the first exon for all 7 loci with spliced long first exon pachytene piRNA clusters.

Supplementary Fig. 6a. THOC1 and THOC2 RIP signals in the first exon and other exons for each of the seven spliced long-first-exon pachytene piRNA clusters.

2.17 The piRNA abundance normalization could be problematic. According to the description, multi-mapping reads were kept for the analysis, but they were excluded during normalization. This should be clarified.

Response:

Thank you for pointing this out. We also considered normalizing piRNA reads to the number of unique-mapping + multiple-mapping reads, but as most of the mapped reads are uniquely mapped to the mouse genome (90.6% in adult testis, 89.7% in pachytene spermatocyte, and 90.0% in round spermatid), normalizing to unique-mapping or normalizing to unique-mapping + multiple-mapping reads results in similar piRNA abundance. We have added this information in Methods.

2.18 The RNA-seq and RIP-seq alignment is described as “allowing up to 2 mismatch and up to 100 mapping locations”. It is generally not advisable to count the same read multiple times. The authors should clarify the mapping parameters. How did this affect the downstream analysis and interpretation?

Response:

We used the default mapping parameters of the STAR algorithm, “allowing up to 2 mismatch and up to 100 mapping locations”. After mapping, we only used uniquely mapped reads to

compute the RNA-seq and RIP-seq signals using HTSeq. We have now clarified these details in Methods.

2.19 Are the number of piRNA cluster loci described in the introduction correct? For instance, the number of pre-pachytene clusters reported in Özata et al (2020) was much higher than 83.

Response:

In Introduction of this manuscript, we referred to the number of annotated human pre-pachytene clusters as “at least 83”. When we defined human piRNA clusters in Özata et al (2020), we indeed found more than 83 pre-pachytene piRNA clusters. However, manual curation of pre-pachytene piRNA clusters is very time consuming and thus we performed manual curation in Özata et al (2020) only for the 83 pre-pachytene piRNA clusters that produced the most abundant piRNAs, and we used these 83 pre-pachytene piRNA clusters in this manuscript.

2.20 “79 paired-end reads” should be “79 nt paired-end reads”

Response:

We have changed “79 paired-end reads” to “79-nt paired-end reads”. Thank you for pointing this out.

2.21 “Trim Galore” should be “Trim Galore!”

Response:

We have changed “Trim Galore” to “Trim Galore!”. Thank you for pointing this out.

2.22 “and 30 display both features” should be “and 3 display both features”

Response:

We double-checked the number and there are indeed 30 pachytene piRNA clusters that are both unspliced or have long-first-exons and have low-CG promoters.

2.23 It would be useful to include a supplementary table listing the refined piRNA loci in mouse and other species.

Response:

You are right. We have added Supplementary Table 4 listing the refined piRNA loci in mouse and other species.

2.24 What does “Note: Mapped data were downloaded directly from the ENCODE portal.” in Table S3 refer to?

Response:

This note was to indicate the ENCODE data used in our analysis. Sorry, it was unclear. Now we have removed this note but instead included in Table S3 the accessions for the ENCODE data that were downloaded from the ENCODE Portal.

Reviewer #3 (Remarks to the Author):

Yu et al. present a systematic evaluation of the sequence features and epigenetic landscapes that distinguish pachytene piRNA precursor gene expression, and targeting of these genes for piRNA production, in adult male testes. It has previously been determined that a small number of piRNA precursor genes are the source of piRNA production in germline cells. However, many questions have remained about how the expression of pachytene piRNA precursors is specified to the male germline, how the cell identifies certain genes for piRNA production, and what keeps these pathways silent in somatic tissues. This study combines re-analysis of a large number of public datasets as well as three newly generated RIP- and ChIP-seq datasets to try to determine whether certain genomic and epigenomic features distinguish piRNA precursors from other germline expressed transcripts (and silencing of these loci in the soma).

Curious features that appear to distinguish pachytene piRNA loci include: high levels of DNA methylation despite relatively low CG dinucleotide content, high levels of 5 hydroxymethylation in addition to CG methylation, abundantly expressed transcripts from genes that are either unspliced or contain very long first exons, and high levels of histone acylation that appears to be broadly distributed around the promoters and across the gene bodies of these regions. It is still unclear which of the epigenetic marks is laid down first, which epigenetic writers lay down these marks, and what proteins recognize these features as piRNA specific. However, this provides some very nice new information about the distinguishing features of piRNA precursors that produce the most abundant piRNAs.

I do have some questions that I think need to be addressed in order to make the story more clear.

Response:

Thank you for your accurate and positive assessment of our work.

3.1 In Figure 1A, in the paragraph starting on Line 218, and in several places through the text, much is made of the high methylation levels of pachytene piRNA gene promoters. Yet, there is very little graphical display of CG methylation levels across these genes. Instead, we get a correlation coefficient and p-value table in Figure 2A (btw, hopefully, multiple hypothesis corrections have been made to this series of correlation p-values – Bonferroni or BH?). Since this is a major conclusion, it seems like this data merits display, preferably in wiggle/profile format. I did see the CG methylation graphs in Fig 5D-E for pachytene spermatocytes and a few somatic tissues. But there seem to be several other tissues with WGBS data in Supplemental table 8, and the discussion of 5mC levels begins long before Fig 5 is introduced. More specifically, since these precursors have the rather uncommon feature of showing both high 5hmC and high 5mC, is there a way of displaying relative levels of the two types of methylation on piRNA precursor promoters versus other genes? Or in the germline versus soma?

Response:

In the original submission, the p-values in Fig. 2a were not corrected for multiple testings. We have now corrected this by reporting the Benjamini-Hochberg adjusted p-values in Fig. 2a. The 5mC level is a percentage based on WGBS data, but the 5hmC signal is computed from 5hMe-DIP data as a fold enrichment. We now show the average profiles of CG methylation and 5hmC level in the “wiggle” format (the newly added Supplementary Fig. 6c and Supplementary Fig. 7 in the revised manuscript, also shown below). In Supplementary Fig. 6c, the 5hmC levels at low-CG genes in pachytene spermatocytes are shown, and 5hmC is specifically enriched in the promoters of low-CG pachytene piRNA clusters. In Supplementary Fig. 7, we compared CG methylation and 5hmC levels of pachytene piRNA clusters between the germline and the soma. Supporting the text in Results, low-CG pachytene piRNA clusters are hypermethylated in both the germline and the soma, but have high 5hmC levels in the germline only (left panels); intermediate-CG pachytene piRNA clusters are hypermethylated in the soma but hypomethylated in the germline (top middle panel), and high-CG pachytene piRNA clusters are hypomethylated in both the germline and the soma (top right panel).

Supplementary Fig. 6c.

The average profiles of 5hmC signals around the transcription start site in pachytene spermatocytes among different types of genes with low-CG promoters.

Supplementary Fig. 7. DNA methylation and 5hmC profiles of low-CG, intermediate-CG and high-CG pachytene piRNA clusters in pachytene spermatocytes and somatic tissues.

3.2 Beginning on line 256, there's discussion of the "broadness" of several epigenetic marks, extending far into the piRNA gene bodies. Yet, if you look at Fig 2D, these marks don't seem predominantly asymmetric with a bias in the direction of transcription. They simply seem to cover a broad region on both sides of the TSS. Moreover, in several of the ChIP heatmaps and wiggle plots, these are cut off a few kb upstream of the TSS, but clearly extend in both directions. Is this primarily due to genes with bidirectional promoters that are driving the upstream enrichments? Or, is this broad enrichment more common across even unidirectionally transcribed piRNA genes? I guess I'm asking if the boundaries of the broad histone patterns correlate with RNA/transcription boundaries? Or if instead, the epigenetic landscapes that define piRNA gene loci occupy a larger surrounding region and potentially influence (or are restricted by) surrounding genes?

Response:

Yes, the broad ChIP-seq signals of histone acylation and BTBD18 (Figs. 2d and 3a, b, Supplementary Fig. 4a, b) are primarily due to the 15 pairs of bidirectional pachytene piRNA clusters (one example pair of bidirectional piRNA clusters is shown in Fig. 3d). For unidirectional pachytene piRNA genes, the ChIP-seq signals of histone acylation and BTBD18 only extend in the transcriptional direction, but not upstream (one example of unidirectional piRNA cluster is shown in Fig. 3c). Unlike ChIP-seq and ATAC-seq signals, THOC1 and THOC2 RIP signals are specific to the genomic strand that is transcribed and hence do not show any signal upstream the TSS (Fig. 4b). We have added a sentence to explain these differences in Results.

3.3 For Fig 4B, I was struck by the wording that describes THOC1/2 binding that seems to end at the first splice site or 3'end (whichever is first). You've put it down as THOC1/2 being loaded before intron removal but after recognition of the 5' splice site. Is the data not also consistent with a scenario that says THOC1/2 could be loaded anytime until the polymerase reaches the first 5' splice site or 3' processing site? Perhaps there's additional arguments or literature on THO complex function that guided your choice of words, but I'm curious about the rationale for this interpretation.

Response:

You are right. We currently do not have sufficient data to distinguish the two possibilities you raise. Current data suggest a co-binding of the THO complex with RNA Pol II, and we speculate that the assembly of a functional splicing complex may block THO binding. We have now changed the sentence from “before intron removal but after recognition of the 5' splice site” to “the THO complex binds to transcripts from the 5'-end to the first 5' splice site”.

3.4 Given the refinements to the piRNA gene annotations described in the methods, which sound very reasonable, I think it would be a nice thing for the field if an updated table of piRNA gene structures was given as an additional supplemental table. Ideally, it would be nice if this were in bed, GTF, or GFF format.

Response:

Following your suggestion, we have now added Supplementary Table 4 listing the refined piRNA clusters in mouse and other species. The content of the table can be saved as a file in the BED12 format.

Reviewer's Table 1. Evolutionary conservation of pachytene piRNA clusters between human and mouse

a. Human syntenic regions of mouse pachytene piRNA clusters and their expression and piRNA production in mouse

b. Mouse syntenic regions of human pachytene piRNA clusters and their expression and piRNA production in human

Reviewer's Table 1a. mouse pachytene piRNA clusters

mouse piRNA cluster	first exon length in mouse	syntenic region in human	first exon length in human	expression level in human (RPKM)	piRNA production in human (ppm)	human piRNA cluster in syntenic region
2-qE1-35981.1	57,935	chr11:45676950-45743767:-	49,707	11.4	37834.7	11-p11-43732
7-qD1-16444.1	56,595	chr15:97271812-97326560:-	21,729	6.1	5188.5	15-q26-7771
4-qC5-17839.1	51,471	chr9:26604680-26644601:-	42,510	3.5	2201.7	9-p21-2544
10-qB4-6488.1	50,297	chr10:70618296-70660553:-	44,236	3.1	611.8	NA
14-qA3-19970.1	48,097	not syntenic	NA	NA	NA	NA
6-qF3-28913.1	45,459	chr12:3543108-3570687:-	24,099	3.4	648.6	12-p13-836
10-qC1-1527.1	42,830	not syntenic	NA	NA	NA	NA
9-qA5.3-24188.1	42,649	chr15:51547208-51595278:-	54,481	10.2	22419.3	pi-CYP19A1
9-qC-31469.1	42,534	chr15:62518890-62533156:+	58,286	10.5	48527.9	15-q22-56093
10-qC1-875.1	42,308	not syntenic	NA	NA	NA	NA
5-qG3-23659.1	41,517	chr13:31910021-31954774:-	36,028	9.2	3486.5	13-q12-4738
7-qD1-19431.1	41,467	chr15:93900261-93965861:-	13,054	9.1	8245.9	15-q26-9530
12-qE-23911.1	41,103	chr14:88579365-88626453:-	27,673	6.6	11657.7	14-q31-15834
7-qD2-24830.1	40,086	chr15:93124043-93163245:+	39,251	5.5	7520.2	15-q26-10588
17-qA3.3-26735.1	39,458	chr6:33860776-33903606:+	25,045	11.3	37412.6	6-p21-43244
18-qE1-36451.1	37,013	chr18:11592000-11670113:-	21,316	17.9	9016.0	18-p11-13059
17-qA3.3-27363.1	36,752	chr6:33816154-33860498:-	24,946	10.9	14640.6	6-p21-16923
5-qF-14224.1	36,736	not syntenic	NA	NA	NA	NA
15-qD1-4001.1	36,647	chr8:125954546-125990180:+	37,258	9.5	206.6	8-q24-335
12-qE-7089.1	35,817	chr14:88629108-88673212:+	30,768	4.4	1193.3	14-q31-1379
7-qD1-9417.1	35,373	chr15:97326725-97358221:+	424	104.5	147.7	pi-SPATA8
15-qE1-1119.1	35,227	chr22:39541983-39586409:+	low expression	0.0	1.5	NA
10-qC1-2617.1	33,188	not syntenic	NA	NA	NA	NA
14-qA3-2286.1	32,151	not syntenic	NA	NA	NA	NA
14-qC1-1261.1	31,826	not syntenic	NA	NA	NA	NA
6-qD1-2831.1	31,711	chr3:129029850-129035958:-	low expression	0.0	3.2	NA
11-qE1-9443.1	29,369	not syntenic	NA	NA	NA	NA
15-qD1-17920.1	27,856	chr8:125908287-125954162:-	49,361	3.8	4025.7	8-q24-4652
9-qA5.3-1495.1	27,692	chr15:51595403-51625392:+	low expression	0.8	41.0	NA
14-qA3-3095.1	27,250	not syntenic	NA	NA	NA	NA
9-qC-10667.1	26,984	chr15:62455997-62516253:-	28,174	7.4	7110.2	15-q22-8218
10-qC1-12816.1	26,646	chr22:24235802-24256140:-	22,891	8.7	5059.7	22-q11-5848
5-qF-4633.1	25,067	not syntenic	NA	NA	NA	NA
17-qE1.1-7037.1	24,946	not syntenic	NA	NA	NA	NA

9-qF4-150.1	24,796	chr3:44328871-44379391:-	low expression	0.0	0.4	NA
15-qE1-8387.1	22,847	chr22:37745030-37767690:-	18,597	25.5	15936.8	pi-ELFN2
15-qD3-14639.1	22,523	not syntenic	NA	NA	NA	NA
5-qF-14508.1	19,640	not syntenic	NA	NA	NA	NA
14-qC1-1010.1	18,400	not syntenic	NA	NA	NA	NA
10-qB5.1-5404.1	18,236	chr10:65733431-65755961:+	low expression	0.1	2.3	NA
6-qF3-8009.1	18,035	chr12:3570744-3594669:+	29,901	4.5	5250.7	12-p13-6069
18-qE1-1295.1	17,791	chr18:11670332-11674862:+	18,594	5.8	1639.7	18-p11-2021
14-qA3-284.1	17,008	not syntenic	NA	NA	NA	NA
10-qC1-117.1	16,543	chr12:106591650-106610083:+	low expression	0.0	0.1	NA
7-qF3-246.1	14,949	chr16:27059051-27075382:+	low expression	0.0	0.0	NA
10-qB5.1-221.1	10,399	chr10:65711959-65733204:-	low expression	0.0	0.2	NA
pi-1700016M24Rik.1	10,179	chr8:143646438-143662062:-	low expression	0.0	0.1	NA
7-qD2-11976.1	9,436	chr15:93106580-93123910:-	9,590	25.5	7232.3	15-q26-13520
17-qC-59.1	8,380	not syntenic	NA	NA	NA	NA
4-qD2.2-2182.1	8,341	chr8:141541822-141561486:+	low expression	0.0	0.3	NA
4-qB3-639.1	7,913	not syntenic	NA	NA	NA	NA
11-qE1-252.1	7,849	chr17:64188049-64197015:+	low expression	0.0	0.4	NA
8-qA4-332.1	6,020	not syntenic	NA	NA	NA	NA
8-qC5-8200.1	4,956	chr16:54753336-54758417:+	low expression	0.0	0.1	NA
1-qD-4525.1	4,722	not syntenic	NA	NA	NA	NA
7-qD1-654.1	4,620	chr15:93965899-93971925:+	420	3.1	309.8	NA
4-qD2.2-349.1	4,517	not syntenic	NA	NA	NA	NA
11-qB1.3-590.1	4,129	not syntenic	NA	NA	NA	NA
6-qC3-100.1	3,716	chr2:74073766-74078671:-	588	1.1	0.0	NA
8-qA2-343.1	3,211	not syntenic	NA	NA	NA	NA
1-qE3-706.1	3,087	not syntenic	NA	NA	NA	NA
3-qA2-617.1	2,859	not syntenic	NA	NA	NA	NA
10-qA3-143.1	2,254	chr6:136610687-136612535:+	1,849	2.2	1.0	NA
6-qC3-2394.1	1,802	chr2:74071906-74081596:-	low expression	0.7	0.0	NA
8-qA4-155.1	1,610	chr1:200182689-200182955:-	low expression	0.0	0.0	NA
6-qC3-6258.1	1,601	not syntenic	NA	NA	NA	NA
2-qF1-2536.1	829	chr2:74058156-74074924:+	low expression	0.6	1.5	NA
8-qC5-2209.1	679	not syntenic	NA	NA	NA	NA
pi-Gm5878.1	504	not syntenic	NA	NA	NA	NA
5-qG2-2301.1	410	not syntenic	NA	NA	NA	NA
17-qC-935.1	338	chr6:40346079-40347618:+	561	40.4	8997.6	pi-TDRG1
pi-Wdfy3.1	336	chr4:85590704-85887816:-	low expression	0.7	4.2	NA
pi-Arhgap20.1	312	chr11:110447088-110583193:-	low expression	0.5	3.0	NA
4-qB3-277.1	306	not syntenic	NA	NA	NA	NA
13-qA3.1-213.1	304	chr6:28132650-28135376:+	40	1.2	0.6	NA
5-qG2-950.1	285	chr7:100916535-100928742:-	low expression	0.0	0.4	NA
19-qC2-1361.1	269	chr10:94176571-94179714:-	3,144	7.6	43.3	NA
9-qA1-178.1	255	chr3:185332351-185348950:-	low expression	0.0	0.1	NA
4-qD3-2082.1	249	chr1:24578618-24581294:-	28	2.8	390.9	1-p36-1107

7-qF3-3125.1	246	chr16:27038409-27058438:-	low expression	0.1	0.2	NA
4-qB3-3994.1	232	chr9:112318858-112324497:-	low expression	0.0	0.0	NA
10-qA3-2592.1	214	not syntenic	NA	NA	NA	NA
10-qC2-545.1	209	chr12:94853953-94868889:+	2,481	1.0	4.4	NA
pi-Tmem194.1	183	chr12:57449424-57482299:-	112	1.5	14.9	NA
13-qA5-703.1	139	not syntenic	NA	NA	NA	NA
17-qA3.3-352.1	125	not syntenic	NA	NA	NA	NA
1-qD-2017.1	123	chr2:239134676-239136553:-	318	3.3	68.3	NA
13-qA5-464.1	120	not syntenic	NA	NA	NA	NA
1-qC1.3-637.1	115	chr2:200741969-200775898:-	low expression	0.7	4.0	NA
13-qA3.1-355.1	108	chr6:24498308-24537580:-	low expression	0.1	2.1	NA
13-qA5-967.1	105	chr9:92686228-92699326:+	low expression	0.0	0.0	NA
2-qG3-1029.1	91	not syntenic	NA	NA	NA	NA
8-qE1-3748.1	87	chr16:71433951-71449910:-	low expression	0.1	24.9	NA
3-qA3-2052.1	77	chr3:181670181-181728524:+	128	13.1	1098.3	3-q26-1568
13-qA5-208.1	76	not syntenic	NA	NA	NA	NA
pi-Cdc42ep3.1	73	chr2:37871987-37898621:-	109	2.5	5.1	NA
13-qB1-1517.1	62	not syntenic	NA	NA	NA	NA
pi-1700006A11Rik.1	55	not syntenic	NA	NA	NA	NA
7-qB5-6255.1	50	chr2:74076373-74081530:-	588	1.1	0.0	NA
11-qE1-3997.1	41	chr17:64705729-64712724:-	low expression	0.0	0.0	NA

Reviewer's Table 1b. human pachytene piRNA clusters

human piRNA cluster	first exon length in human	syntenic region in mouse	first exon length in mouse	expression level in mouse (RPKM)	piRNA production in mouse (ppm)	mouse piRNA cluster in syntenic region
4-p16-5577	72,820	not syntenic	NA	NA	NA	NA
15-q22-56093	58,286	not syntenic	NA	NA	NA	NA
pi-CYP19A1	54,481	chr9:54201415-54249844:-	42,649	5.7	25871.9	9-qA5.3-24188.1
11-p11-43732	49,707	chr2:92541187-92592844:+	57,935	3.5	42997.7	2-qE1-35981.1
8-q24-4652	49,361	chr15:59240123-59275224:-	27,856	7.0	21628.4	15-qD1-17920.1
10-q23-411	43,235	chr19:34989648-35020177:-	NA	0.0	0.0	
9-p21-2544	42,510	chr4:94281319-94331618:-	51,471	3.0	18843.0	4-qC5-17839.1
15-q26-10588	39,251	chr7:73773862-73816453:-	40,086	4.1	30237.9	7-qD2-24830.1
2-q13-9453	38,321	not syntenic	NA	NA	NA	NA
6-p21-438	37,831	chr14:20083070-20107304:+	NA	0.2	0.0	
8-q24-335	37,258	chr15:59278207-59283530:+	36,647	2.5	5472.9	15-qD1-4001.1
13-q12-4738	36,028	chr5:149804683-149825409:-	41,517	5.2	27634.0	5-qG3-23659.1
17-p11-4442	33,082	not syntenic	NA	NA	NA	NA
17-p11-4161	30,970	chr11:116232350-116236474:+	NA	0.3	0.0	
14-q31-1379	30,768	chr12:98420587-98439969:+	35,817	7.5	8806.4	12-qE-7089.1
12-q24-1580	30,709	chr5:113340222-113347971:+	36,736	11.1	19084.9	5-qF-14224.1
12-p13-6069	29,901	chr6:127768903-127796380:-	18,035	13.8	9650.7	6-qF3-8009.1
12-p13-1820	29,010	not syntenic	NA	NA	NA	NA
15-q22-8218	28,174	chr9:67734068-67742020:+	26,984	5.0	12980.8	9-qC-10667.1
9-q21-769	27,828	chr19:14316579-14343412:-	NA	0.0	0.1	
14-q31-15834	27,673	chr12:98393023-98418421:-	41,103	8.0	25461.1	12-qE-23911.1
3-p25-2999	27,066	chr6:115894147-115904556:-	NA	0.0	0.7	
9-q21-8235	25,452	not syntenic	NA	NA	NA	NA
6-p21-43244	25,045	chr17:27325406-27348670:+	39,458	7.7	43649.9	17-qA3.3-26735.1
6-p21-16923	24,946	chr17:27310318-27324774:-	36,752	4.9	41137.9	17-qA3.3-27363.1
12-p13-836	24,099	chr6:127796460-127823093:+	45,459	2.5	32436.7	6-qF3-28913.1
15-q26-361	24,009	chr7:75114609-75134167:+	NA	0.1	0.0	
12-p13-6740	23,809	chr12:89551706-89569053:+	NA	0.0	0.0	
22-q11-5848	22,891	chr10:75837397-75869855:+	26,646	8.7	15644.6	10-qC1-12816.1
15-q26-7771	21,729	chr7:69909781-69934996:+	56,595	2.5	17836.7	7-qD1-16444.1
18-p11-13059	21,316	chr18:67045058-67066510:-	37,013	5.3	40421.0	18-qE1-36451.1
3-q22-367	20,871	not syntenic	NA	NA	NA	NA
10-q22-16903	19,914	not syntenic	NA	NA	NA	NA
pi-ELFN2	18,597	chr15:78661420-78670619:-	22,847	7.7	10967.1	15-qE1-8387.1
18-p11-2021	18,594	chr18:67067131-67073374:+	17,791	9.4	1777.3	18-qE1-1295.1
7-q31-418	15,700	not syntenic	NA	NA	NA	NA
17-q12-687	13,767	not syntenic	NA	NA	NA	NA
15-q26-9530	13,054	chr7:73105368-73116876:+	41,467	6.5	21730.6	7-qD1-19431.1
16-p12-3787	11,779	not syntenic	NA	NA	NA	NA
15-q26-13520	9,590	chr7:73816505-73834434:+	9,436	17.0	14263.0	7-qD2-11976.1
pi-TRIM14	9,130	chr4:46505079-46536082:-	NA	0.7	56.9	

pi-DGCR9	7,857	not syntenic	NA	NA	NA	NA
pi-NPAP1	7,526	chr5:135380910-135382138:-	NA	0.0	0.3	
21-q22-4766	6,814	not syntenic	NA	NA	NA	NA
19-q13-2441	6,283	not syntenic	NA	NA	NA	NA
15-q26-74991	5,239	not syntenic	NA	NA	NA	NA
10-q22-295	4,134	chr14:37790667-37794106:+	NA	0.0	0.0	
7-p11-9381	3,598	chr13:35901788-35905752:-	NA	0.5	0.0	
5-q35-131	3,398	not syntenic	NA	NA	NA	NA
9-q22-2190	1,752	not syntenic	NA	NA	NA	NA
19-p13-4412	1,568	not syntenic	NA	NA	NA	NA
10-p11-10777	1,454	chr18:3648230-3652874:-	NA	0.0	0.0	
pi-FAM120AOS	1,446	chr13:48965841-48974265:+	NA	0.7	0.2	
17-q11-3352	1,289	chr11:102515722-102538458:-	NA	0.6	0.3	
9-q22-405	1,259	chr13:52488966-52492003:-	NA	0.0	0.0	
19-q13-7835	910	not syntenic	NA	NA	NA	NA
10-q23-715	795	chr19:7275407-7290777:+	NA	0.0	0.0	
16-p12-3782	710	not syntenic	NA	NA	NA	NA
9-q31-7934	591	not syntenic	NA	NA	NA	NA
pi-TDRG1	561	chr17:49112263-49113514:-	338	130.0	1253.4	17-qC-935.1
pi-ANKRD36B	441	not syntenic	NA	NA	NA	NA
pi-SPATA8	424	chr7:69907576-69909667:-	35,373	2.3	13211.0	7-qD1-9417.1
3-q29-236	405	chr2:68570228-68578548:+	462	1.8	0.2	
10-q11-242	359	chr6:116629005-116629912:+	NA	0.5	0.0	
2-q21-678	351	not syntenic	NA	NA	NA	NA
2-q37-1034	318	chr1:91403917-91405721:-	123	34.7	2192.5	1-qD-2017.1
20-p12-2902	316	chr2:132490661-132492038:+	329	4.2	0.3	
12-q13-1035	308	chr10:126906022-126908046:+	NA	0.7	0.0	
pi-UNC119B	284	chr5:115122560-115134949:-	308	2.0	211.4	
pi-NEU3	250	chr7:99810512-99828308:-	NA	0.2	1.5	
12-q23-3486	232	chr2:32288163-32306710:+	86	3.0	8.8	
19-q13-4180	211	not syntenic	NA	NA	NA	NA
pi-SAYSD1	189	not syntenic	NA	NA	NA	NA
19-p13-25455	175	chr13:113616800-113618397:+	NA	0.0	0.0	
1-q25-2976	172	not syntenic	NA	NA	NA	NA
6-p22-5808	165	not syntenic	NA	NA	NA	NA
11-p11-1759	150	chr2:92540470-92541001:-	89	8.0	25.9	
19-q13-13274	145	not syntenic	NA	NA	NA	NA
5-q35-1815	141	not syntenic	NA	NA	NA	NA
19-p13-6652	138	chr13:113616800-113618461:+	NA	2.7	0.0	
6-p24-803	135	chr13:41220754-41239587:+	NA	0.0	0.4	
3-q26-1568	128	chr3:34826590-34879241:+	77	11.7	2141.5	3-qA3-2052.1
15-q26-10235	114	not syntenic	NA	NA	NA	NA
pi-GOLGA2	97	chr2:32288298-32307911:+	86	3.5	42.9	
18-p11-676	87	chr17:67835166-67836469:+	NA	0.1	0.0	
pi-TMEM99	84	chr11:99386227-99396486:+	NA	0.7	0.3	

9-q31-7830	81	chr4:42956484-42958017:+	190	6.9	0.0	
5-q35-600	56	chr18:67812041-67854235:+	79	1.2	0.6	
1-p36-1107	28	chr4:135558049-135629129:+	249	56.2	2717.7	4-qD3-2082.1

REVIEWER COMMENTS

Reviewer #1 (Remarks to the Author):

Although I am glad to see in this revision the authors temper the claims that were too strong in the original submission, I am still not satisfied with the revision in not properly addressing the main issue about applying their analysis more thoroughly to other piRNA clusters differentially expressed in just mouse or just human or another specie, regardless of deeper orthology or synteny with the other mammals of cow, rat or monkey. I would like to know if the features of long exons AND Epigenetic marks specially track with the mouse-specific or human-specific loci (or other species-specific loci) that happens to be expressing high levels of piRNAs in one species but not in the ortholog of the other specie.

The Reviewer Figure 1 does not exactly address the question I raised. It does compare one syntenically conserved piRNA cluster highly expressed for piRNAs in mouse while being much lower expressed in piRNAs human (and the transcript levels do not correlate with piRNA levels) But there is also significant piRNA expression in rat, monkey, marmoset and cow, but the human and rhesus piRNA cluster has undergone a genomic strand directionality flip versus the other animals. Only the transcript patterns are shown but no other epigenetic marks are shown, like DNA methylation, histone methylation, transcription factor binding (A-Myb or BTBD18), or THOC association.

The lack of other epigenetic mark data from other animal testes is a big issue here which is not irrelevant since transcriptome data is being used in a comparative genomics analysis in Figure 6, and one histone mark H3K27Ac Chip is shown for Rhesus. Surely there has to be more than 1 piRNA cluster that is differentially expressed just in mouse or just in rhesus (or just in human) that exists as orthologs or homologs between the species that can then be scrutinized to see the exon length And Epigenetic mark tracks with the specific specie making a lot of piRNAs from those clusters? Aren't there public datasets of histone marks from human testes that can be data-mined?

Is there a similar matrix of Epigenetic Marks correlating with high piRNA production for a more limited set of species like the Exon analysis of Fig 6D? And of the piRNA clusters in Fig 6A and 6D who have short exons but also relatively high piRNA expression levels, how does your study explain these features and fits them into the paper title? There is not an insignificant number of Short Exon Pachytene piRNA clusters I notice more obviously now in Fig. 6A. If there was an additional combination of Epigenetic Marks (the CG methylation and specific histone, transcription factor, Thoc marks, etc) that are on these Short Exon piRNA clusters but Absent from protein coding genes, would we see these marks track on the species specific examples? This is the type of validation I was asking for originally.

BTW, Fig 3C, the BTBD18 label has a typo.

Reviewer #2 (Remarks to the Author):

Dear Editor,

in this revised manuscript, the authors have performed additional analyses and included extra details, thus addressing most of my comments. I do feel that this improved manuscript version is ready for publication, but I found a few more minor points that the authors might want to take into consideration:

1) The labels of the figures should be updated throughout. For example, in Fig 3a “piRNA-producing genes” and “pachytene piRNA genes” in Fig 4B should be changed to “piRNA clusters”. Same in various panels of Fig 5.

2) Supplementary Table 3: The authors added accession numbers for the GEO/ENCODE data which will be helpful to the readers. However, some of those samples are listed to have 100% mapped reads (e.g. replicate 2 of ENCSR000CEG) – this looks odd, is this a typo?

3) Typos:

- Line 998: “ENSEMBLE” should be “Ensembl”

- Supplementary Table 3: “repicates” in sheet “small RNA-seq” and “repplicates” in sheet “Bisulphite-seq” should both be “replicates”.

Reviewer #3 (Remarks to the Author):

The authors have addressed all of my concerns.

RESPONSE TO REVIEWERS' COMMENTS

Reviewer #1 (Remarks to the Author):

1.1 Although I am glad to see in this revision the authors temper the claims that were too strong in the original submission, I am still not satisfied with the revision in not properly addressing the main issue about applying their analysis more thoroughly to other piRNA clusters differentially expressed in just mouse or just human or another specie, regardless of deeper orthology or synteny with the other mammals of cow, rat or monkey. I would like to know if the features of long exons AND Epigenetic marks specially track with the mouse-specific or human-specific loci (or other species-specific loci) that happens to be expressing high levels of piRNAs in one species but not in the ortholog of the other specie.

Response:

First, human-specific and mouse-specific piRNA-producing loci are highly unlikely to have any function in vivo. By definition, such species-specific loci cannot have orthologs in other species. Moreover, most conserved piRNA clusters in mouse have no phenotype when individually deleted.

Second, most of the epigenetic data (histone acylation, DNA methylation, BTBD18 binding) available for mouse testis are not available for post-pubertal testes from human, rhesus, marmoset, cow, rat, opossum, or platypus, except for the H3K27ac ChIP-seq data in rhesus, which we generated for this study, and the human and rhesus A-MYB ChIP-seq data from our previously published work (Özata et al., 2020, *Nature Ecology & Evolution* 4: 156–68).

Third, we searched thoroughly for all pachytene piRNA genes in mouse and human. Unfortunately, evolution did not perform the experiments to test our hypothesis. Except for the one example in Reviewer's Figure 1, we found no other pachytene piRNA-producing loci with a long first exon in one species that both produced a short first-exon transcript and retained high levels of piRNA production in another species.

1.2 The Reviewer Figure 1 does not exactly address the question I raised. It does compare one syntenically conserved piRNA cluster highly expressed for piRNAs in mouse while being much lower expressed in piRNAs human (and the transcript levels do not correlate with piRNA levels) But there is also significant piRNA expression in rat, monkey, marmoset and cow, but the human and rhesus piRNA cluster has undergone a genomic strand directionality flip versus the other animals. Only the transcript patterns are shown but no other epigenetic marks are shown, like DNA methylation, histone methylation, transcription factor binding (A-Myb or BTBD18), or THOC association.

Response:

As described in our point 1.1 above, we performed a thorough comparison between the piRNA clusters in human and mouse (Reviewer Table 1 of our first revision) and, indeed, found just one piRNA cluster (shown in Reviewer Figure 1 of our first revision) that

expresses a long-first-exon transcript in the mouse but a short-first-exon transcript in humans. In the rat, this piRNA cluster also produces a long-first-exon transcript and makes a comparable amount of piRNAs as mouse. In rhesus, marmoset and cow, the orthologous loci of this piRNA cluster produce short-first-exon transcripts and make fewer piRNAs than in mouse and rat.

As discussed above, most of the epigenetic data available for mice (e.g., histone acylation, DNA methylation, BTBD18 binding) do not exist for adult testes from human, rhesus, marmoset, cow, rat, opossum, or platypus. We believe that asking us to generate such data is unreasonable, given that it could easily take years to accrue sufficient fresh post-pubertal human or monkey testis tissue.

1.3 The lack of other epigenetic mark data from other animal testes is a big issue here which is not irrelevant since transcriptome data is being used in a comparative genomics analysis in Figure 6, and one histone mark H3K27Ac Chip is shown for Rhesus. Surely there has to be more than 1 piRNA cluster that is differentially expressed just in mouse or just in rhesus (or just in human) that exists as orthologs or homologs between the species that can then be scrutinized to see the exon length and Epigenetic mark tracks with the specific specie making a lot of piRNAs from those clusters?

Response:

As described in our manuscript, pachytene piRNA clusters fall into two categories: (1) ancestral clusters whose piRNA production levels correlate with long first exons, BTBD-binding, and histone acylation; and (2) young clusters whose piRNA production is governed by other mechanisms that remain to be fully studied. Clusters in this second class are often confined to individual or closely related species, are unlikely to have a biological function, and are outside the scope of our study.

As described in our point 1.1, above, except for the one example we showed in Reviewer's Figure 1, we did not find another pachytene piRNA gene that both produced a long first-exon transcript and produced piRNAs in mouse or humans, but a short first-exon transcript in rat, rhesus, marmoset and cow.

Aren't there public datasets of histone marks from human testes that can be data-mined?

Response:

Such datasets do not, to the best of our knowledge, exist. In our previous study (Özata et al., 2020, *Nature Ecology & Evolution* 4: 156–68), it took several years to obtain sufficient *fresh, unfixed*, human testis material to do just a few experiments.

1.4 Is there a similar matrix of Epigenetic Marks correlating with high piRNA production for a more limited set of species like the Exon analysis of Fig 6D? And of the piRNA clusters in Fig 6A and 6D who have short exons but also relatively high piRNA expression levels, how does your study explain these features and fits them into the paper title? There is not an insignificant number of

Short Exon Pachytene piRNA clusters I notice more obviously now in Fig. 6A. If there was an additional combination of Epigenetic Marks (the CG methylation and specific histone, transcription factor, Thoc marks, etc.) that are on these Short Exon piRNA clusters but Absent from protein coding genes, would we see these marks track on the species specific examples? This is the type of validation I was asking for originally.

Response:

Unfortunately, to the best of our knowledge, there is no data on epigenetic marks in mature testis of other species.

Among the 100 mouse pachytene piRNA clusters, 53 have short first exons (as the reviewer has noticed in Fig. 6A). These short-first-exon piRNA clusters make fewer piRNAs than the 47 long-first-exon pachytene piRNA clusters, as shown in Fig. 6A and 6D (in pink and red, respectively). Unfortunately, we have not deduced a specific combination of epigenetic marks that can distinguish short-first-exon piRNA clusters from protein-coding genes (many of which also produce piRNAs before and after puberty). The piRNAs produced from these short-first-exon piRNA clusters are evolutionarily younger and we suspect that they are less likely to be functional than the piRNAs produced from the long-first-exon piRNA clusters, just like the abundant piRNAs produced in pre-pachytene spermatocytes from the 3'UTRs of pre-mRNAs.

The Reviewer asks, “*If there was an additional combination of Epigenetic Marks (the CG methylation and specific histone, transcription factor, Thoc marks, etc.) that are on these Short Exon piRNA clusters but Absent from protein coding genes.... This is the type of validation I was asking for originally.*” The Reviewer’s is most definitely not asking for validation of our main findings—i.e., first-exon length and specific epigenetic signatures are associated with the long-first-exon group of piRNA clusters. Instead, the Reviewer is asking us to make additional discoveries on the remaining group of short-first-exon piRNA clusters. Not only are these outside the scope of our study (see above), but if we knew what features of the short-first-exon piRNA clusters impelled them to make piRNAs, we would have highlighted our discovery in the original manuscript.

1.5 BTW, Fig 3C, the BTBD18 label has a typo.

Response:

We have fixed the typo.

Reviewer #2 (Remarks to the Author):

In this revised manuscript, the authors have performed additional analyses and included extra details, thus addressing most of my comments. I do feel that this improved manuscript version is ready for publication, but I found a few more minor points that the authors might want to take into consideration.

Response:

Thank you.

2.1 The labels of the figures should be updated throughout. For example, in Fig 3a “piRNA-producing genes” and “pachytene piRNA genes” in Fig 4B should be changed to “piRNA clusters”. Same in various panels of Fig 5.

Response:

Thank you. We have made these changes.

2.2 Supplementary Table 3: The authors added accession numbers for the GEO/ENCODE data which will be helpful to the readers. However, some of those samples are listed to have 100% mapped reads (e.g. replicate 2 of ENCSR000CEG) – this looks odd, is this a typo?

Response:

The BAM files that we downloaded from the ENCODE portal only contain mapped reads. We have changed 100% to N/A for these datasets and added a note indicating that we started with BAM files for these datasets.

2.3 Typos:

- Line 998: “ENSEMBLE” should be “Ensembl”
- Supplementary Table 3: “repicates” in sheet “small RNA-seq” and “reppicates” in sheet “Bisulphite-seq” should both be “replicates”.

Response:

Thank you. The typos have been corrected.